# Highly accurate carbohydrate-binding site prediction with DeepGlycanSite

Xinheng He [1,2,8], Lifen Zhao[1,8], Yinping Tian[1,8], Rui Li[1,3], Qinyu Chu[4], Zhiyong Gu[4], Mingyue Zheng [1,2,4], Yusong Wang[5], Shaoning Li[6], Hualiang Jiang [1,2,4,7], Yi Jiang [7], Liuqing Wen [1,2] ✉, Dingyan Wang [7] ✉ & Xi Cheng [1,2,4] ✉

As the most abundant organic substances in nature, carbohydrates are essential for life. Understanding how carbohydrates regulate proteins in the physiological and pathological processes presents opportunities to address crucial biological problems and develop new therapeutics. However, the diversity and complexity of carbohydrates pose a challenge in experimentally identifying the sites where carbohydrates bind to and act on proteins. Here, we introduce a deep learning model, DeepGlycanSite, capable of accurately predicting carbohydrate-binding sites on a given protein structure. Incorporating geometric and evolutionary features of proteins into a deep equivariant graph neural network with the transformer architecture, DeepGlycanSite remarkably outperforms previous state-of-the-art methods and effectively predicts binding sites for diverse carbohydrates. Integrating with a mutagenesis study, DeepGlycanSite reveals the guanosine-5′-diphosphate-sugar-recognition site of an important G-protein coupled receptor. These findings demonstrate DeepGlycanSite is invaluable for carbohydrate-binding site prediction and could provide insights into molecular mechanisms underlying carbohydrate-regulation of therapeutically important proteins.

Carbohydrates generally cover living cells in all organisms[1]. Carbohydrates interact with diverse protein families, including lectins, antibodies, enzymes and transporters[1,2], to modulate various biological processes, including immune response, cell differentiation and neural development[3]. Therefore, understanding the carbohydrate-protein interaction offers the essential basis for carbohydrate drug development[2,3]. Previous studies have shown that carbohydrates act as ligands of sialidase to mediate virion release[4,5]. This finding has been converted into the discovery of the well-known medications zanamivir and oseltamivir[6]. There are also two types of oral glucose-based antidiabetic drugs, which are inhibitors of α-glucosidase and sodium-

glucose cotransporter type 2 (SGLT2)[7]. Although substantial progress has been made in targeting carbohydrate-binding proteins, it is challenging to achieve the desired specificity of carbohydrates on the therapeutic target[8]. The lack of understanding regarding how carbohydrates bind to and act on the protein therapeutic targets has hampered the translation of carbohydrates to the clinic[9].

It is difficult to obtain experimental data about carbohydrate-protein interactions, due to the structural diversity of carbohydrates. For example, the most widely used structure determination techniques of glycoscience research, i.e., nuclear magnetic resonance and X-ray crystallography, require pure and stable molecules with

[1]State Key Laboratory of Drug Research and State Key Laboratory of Chemical Biology, Carbohydrate-Based Drug Research Center, Shanghai Institute of Materia Medica, Chinese Academy of Sciences, Shanghai, China. [2]University of Chinese Academy of Sciences, Beijing, China. [3]School of Pharmacy, China Pharmaceutical University, Nanjing, China. [4]School of Pharmaceutical Science and Technology, Hangzhou Institute of Advanced Study, Hangzhou, China. [5]National Key Laboratory of Human-Machine Hybrid Augmented Intelligence, National Engineering Research Center for Visual Information and Applications, and Institute of Artificial Intelligence and Robotics, Xi'an Jiaotong University, Xi'an, China. [6]Department of Computer Science and Engineering, The Chinese University of Hong Kong, Hong Kong, China. [7]Lingang Laboratory, Shanghai, China. [8]These authors contributed equally: Xinheng He, Lifen Zhao, Yinping Tian. ✉e-mail: lwen@simm.ac.cn; wangdy@lglab.ac.cn; xicheng@simm.ac.cn

detectable sizes[10]. The small carbohydrates, such as glucose (Glc) with a low molecular mass of less than 200 Da (Fig. 1a), have few atoms to be detected in the structure determination. The elaborately branched long-chain carbohydrates, such as oligosaccharides with a molecular mass of more than 1000 Da (Fig. 1a), may involve multiple conformational states that lead to heterogeneity. In both cases, the carbohydrate-binding residues of proteins cannot be clearly defined from a structural perspective[10]. Hence, the development of a reliable carbohydrate-binding site predictor is paramount in uncovering the carbohydrate-protein interactions.

However, the prediction of the carbohydrate-binding sites remains challenging. One major obstacle is the complexity of carbohydrate structures. Carbohydrates are composed of various monosaccharides as building blocks[1] (Fig. 1a). These monosaccharide building blocks are joined at one of several positions around the sugar ring, with α- or β-stereochemistry at the anomeric carbon, to form disaccharides (with two monosaccharides) or

oligosaccharides (with at least three monosaccharides). There are numerous ways of linking monosaccharides in their ring forms through glycosidic bonds. In addition, mono-, di- or oligosaccharides can be conjugated to other molecules (e.g., glycolipids) or included as moieties of metabolites (e.g., nucleotides). The complexity is further enhanced by modifications, such as sulfation, methylation, phosphorylation, acetylation and O-acylation[3]. Given the immense complexity of carbohydrates, carbohydrate-binding sites vary substantially in their size and shape (Fig. 1b). Most traditional ligand-binding site prediction methods are tailored for compounds with typical small molecule sizes and, therefore, not suited for carbohydrate-binding site prediction[11–13]. Few methods have been developed specifically for predicting carbohydrate-binding sites. Furthermore, due to a lack of carbohydrate-protein complex structures at atomic resolution, researchers employed the supporting vector machine (SVM) to learn from small datasets of limited samples and developed carbohydrate-binding site predictors with

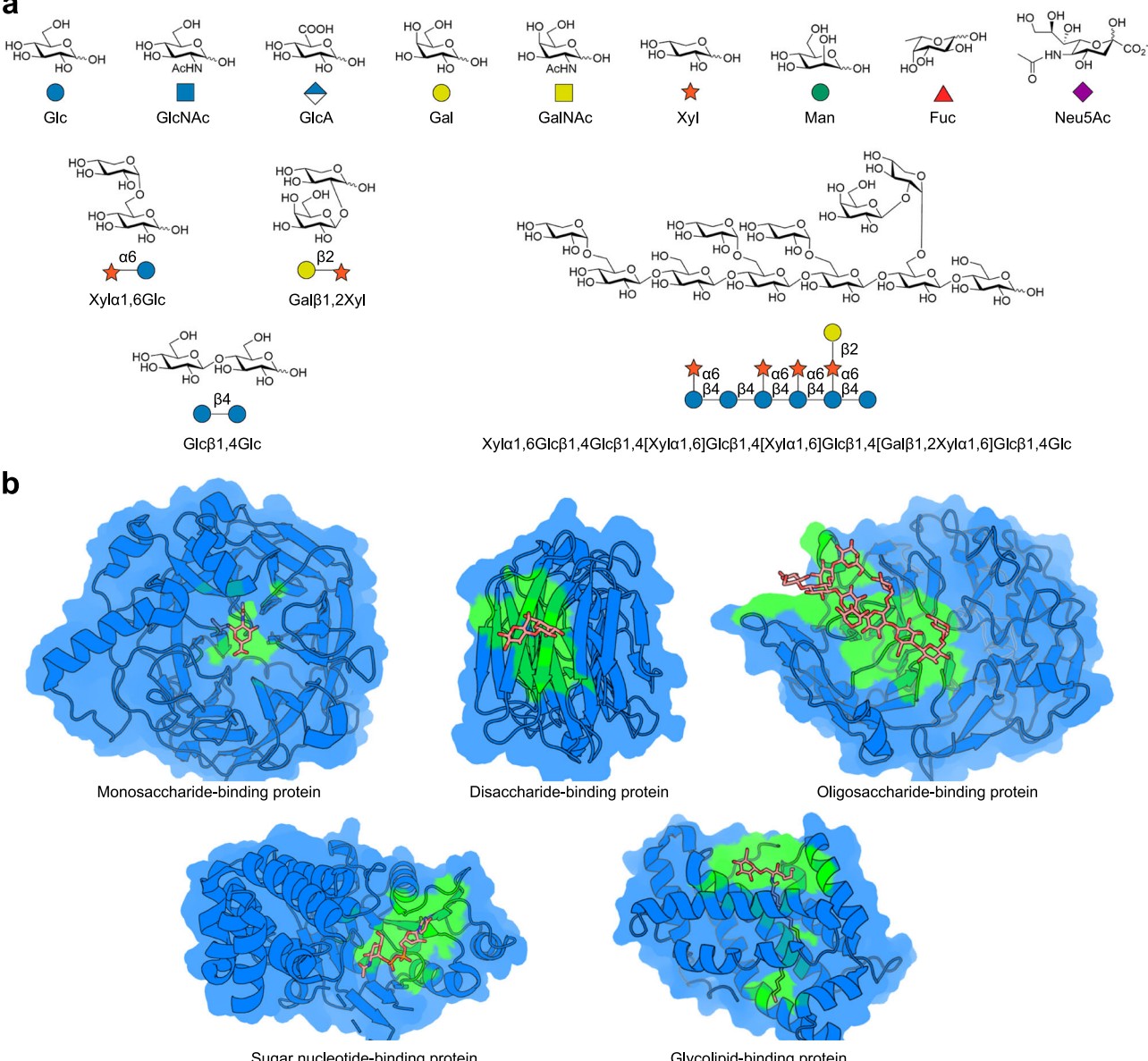

**Fig. 1 | Complexity of carbohydrates and diversity of carbohydrate-binding sites. a** Typical carbohydrates that are pictorially represented by specific symbols from the Symbol Nomenclature of Glycans (SNFG). **b** Representative carbohydrate-binding protein structures showing monosaccharide-, disaccharide-, oligosaccharide-, sugar nucleotide- and glycolipid-binding sites (PDB codes: 1E8U, 4FQZ, 6MGL, 6H21 and 2BV7). Carbohydrates are displayed as sticks. Proteins are shown in cartoon and surface depict. Carbohydrate-binding sites are colored green.

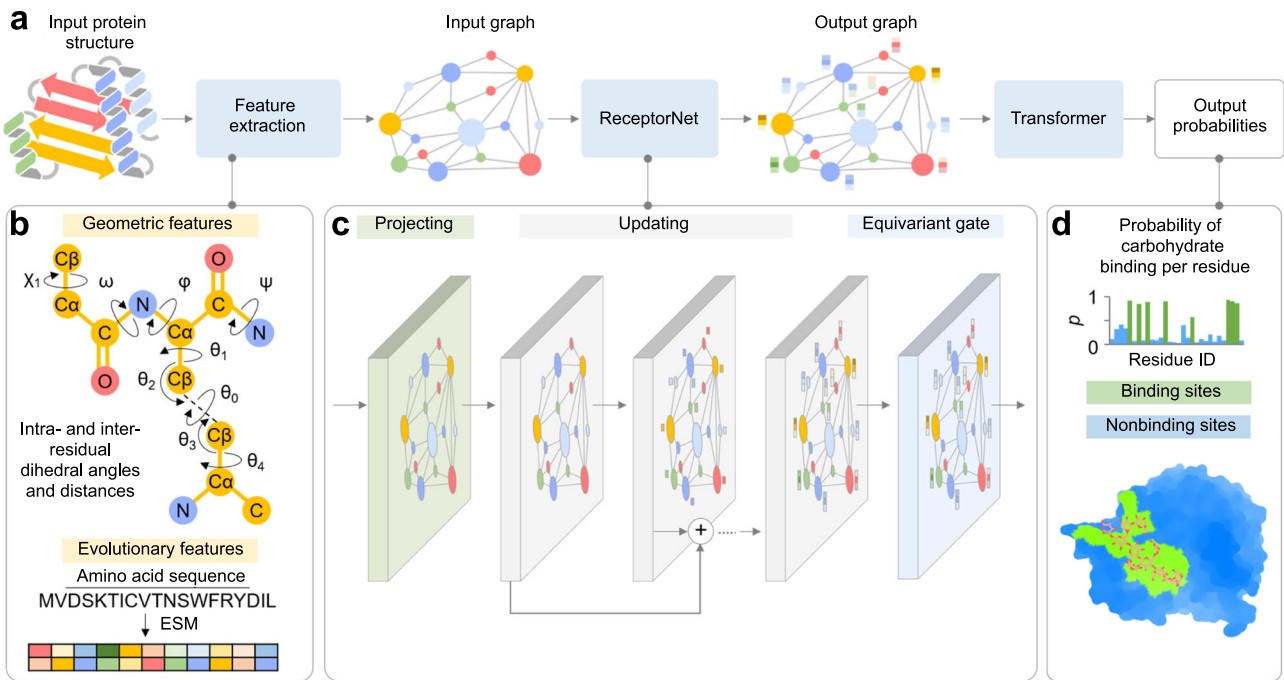

**Fig. 2 | Overview of DeepGlycanSite. a** Model architecture showing the information flow among the various components described in this paper. **b** Geometric and evolutionary features employed in our model. The scheme shows nine dihedral angles as geometric features of residues. **c** Deep equivariant graph neural network trained to predict the carbohydrate-binding residues. **d** Example of output probability of carbohydrate binding per residue. To show the location of the predicted sites, the predicted binding (green) and nonbinding (blue) sites were mapped on the given protein structure. A carbohydrate is shown as sticks to indicate the true carbohydrate-binding site.

modest performance[14–16]. But this situation has been changed recently. Continuous efforts of more than ten years have accumulated plenty of high-quality data for carbohydrates and their target proteins. Protein Data Bank (PDB) and open glycoinformatics resources[9,17–22] have curated more than 19,000 carbohydrate-protein complex structures. The accumulation of high-resolution structural information enables the development of a powerful artificial intelligence (AI)-based method for carbohydrate-binding site prediction.

In this study, we introduce DeepGlycanSite, a deep equivariant graph neural network (EGNN) model capable of accurately predicting carbohydrate-binding sites with the target protein structure. We exploit geometric features, such as intra- and inter-residue orientations and distances, as well as evolutionary information to present proteins in graph representations at the residual level in DeepGlycanSite. Transformer blocks with the self-attention mechanism[23] are incorporated to enhance feature extraction and complex relationship discovery. On the independent testing set involving more than one hundred unique carbohydrate-binding proteins, we compare DeepGlycanSite with the state-of-the-art computational methods. Given the chemical structure of a query carbohydrate, DeepGlycanSite can also predict its specific binding site. We further explored the application of DeepGlycanSite to a functionally important G-protein coupled receptor (GPCR) and then validated it in experimental assays.

## Results

### The DeepGlycanSite network

As shown in Fig. 2, DeepGlycanSite takes a protein structure as the input, and outputs the carbohydrate-binding probabilities of all protein residues. DeepGlycanSite consists of a feature extraction module, a feature fusion module called ReceptorNet and a transformer-based readout module (Fig. 2a). For feature extraction, the atomic coordinates of protein are converted into the intra- and inter-residual geometric features, including the distances and dihedral angles (Fig. 2b). The primary sequence of the protein is employed to generate

evolutionary features using ESM-2 model[24] (Fig. 2b). All features in graph representations are then fed into the ReceptorNet for hierarchical feature fusion, consisting of a projecting block, enhanced vector-scalar interactive updating units (Supplementary Fig. 1a) and an equivariant gate (Fig. 2c). The transformer-based readout module integrates output graph from ReceptorNet to calculate the carbohydrate-binding probability of each residue (Fig. 2d).

To train and test DeepGlycanSite, we curated a large carbohydrate-protein complex dataset, involving ~8100 proteins and more than 1700 carbohydrates (Supplementary Fig. 2). In general, carbohydrate binding specificity could result from limited amino acid changes within a common protein fold[25,26]. In addition, the same protein domain could bind to different carbohydrates[27–29]. Therefore, it is important to include homologous proteins binding to different carbohydrates. To reduce possible bias toward some popular proteins and consider more protein-carbohydrate interactions, we only excluded any instance of the same site binding to the same carbohydrate for the training set. We constructed a training test of 12,507 complex samples. For the testing set, any protein with more than 95% sequence identity to the training set was excluded. We further controlled protein sequence identity of 30% within the testing set to construct an independent dataset of 145 proteins (T145). Notably, we found the proteins have more than one carbohydrate-binding pocket in 15% of the training set and 20% of the testing set. Due to the relatively mild binding affinity, it is sometimes necessary for carbohydrates to bind to multiple sites (or domains) of a protein to achieve their bioactivities[30,31]. We split the T145 into two sets TM29 and TS116. TM29 includes 29 proteins with multiple carbohydrate-binding pockets, while TS116 contains 116 samples in which each protein has only one carbohydrate-binding pocket. A protein may bind to distinct carbohydrates at different sites. We also constructed a testing set of 175 complex samples for these cases (TM175). To assess DeepGlycanSite with various (experimental and predicted) structures of the same protein, we

**Table 1 | Comparing DeepGlycanSite with previous binding site predictors on the independent dataset T145**

| Method | MCC | Precision | Balanced accuracy |
|---|---|---|---|
| StackCBPred | 0.018 ± 0.087*** | 0.052 ± 0.034*** | 0.525 ± 0.100*** |
| Fpocket | 0.191 ± 0.324*** | 0.194 ± 0.278*** | 0.617 ± 0.197*** |
| SiteMap | 0.227 ± 0.400*** | 0.201 ± 0.208*** | 0.717 ± 0.219*** |
| DeepPocket | 0.288 ± 0.479*** | 0.292 ± 0.238*** | 0.760 ± 0.210* |
| PeSTo | 0.336 ± 0.302*** | 0.235 ± 0.167*** | 0.815 ± 0.164 |
| DeepGlycanSite | 0.625 ± 0.292 | 0.631 ± 0.306 | 0.829 ± 0.156 |

Data represent means ± standard deviation. The two-tailed Mann-Whitney U test is used to determine the statistical difference between DeepGlycanSite and an alternative method.
* indicates $P$ is less than 0.05, *** indicates $P$ is less than 0.001. From top to bottom and left to right, the $P$ values of the significantly different groups are 4.5E-36, 1.3E-21, 1.7E-20, 2.2E-13, 4.1E-18, 3.9E-32, 1.7E-23, 4.3E-26, 2.0E-19, 2.3E-24, 2.5E-34, 2.0E-16, 3.5E-5 and 4.0E-2.

excluded proteins of T145 with more than 25% homology from the training set to construct an independent testing set T59. Then, we employed AlphaFold2[32] and AlphaFold2 Multimer[33] to predict protein structures based on the protein sequences of T59. The top five ranked conformation models for each protein were selected to construct an independent dataset T59$_{AF2}$, consisting of 59 unique proteins and 295 structure models. Further details are provided in Methods.

## Carbohydrate-binding site prediction with protein structure

We compared DeepGlycanSite with StackCBPred[14], DeepPocket[34], PeSTo[35], Fpocket[13] and SiteMap[36] in carbohydrate-binding site prediction on the testing set T145. StackCBPred is the only accessible carbohydrate-binding site predictor. DeepPocket and PeSTo are state-of-the-art deep-learning methods for ligand-binding interface prediction, and Fpocket[13] and SiteMap[36] are long-standing methods for ligand-binding site prediction. We used Matthews correlation coefficient (MCC), precision and balanced accuracy to evaluate the model performance as previous studies[14–16]. Large values of these measures indicate good performance. As shown in Table 1, DeepGlycanSite remarkably outperformed the alternative methods in all measures (with an average MCC of 0.625, precision of 0.631 and balanced accuracy of 0.829). Its robustness was affirmed in the five-fold cross-validation (CV) (Supplementary Table 1). All alternative methods have small average MCC (less than 0.350) and precision (less than 0.300), suggesting their inadequate capabilities in carbohydrate-binding site predictions.

We further analyzed the results for different carbohydrate-binding site classes. For monosaccharide- or disaccharide-binding site prediction, DeepGlycanSite showed outstanding performance with the average MCC and precision, more than twice those of the alternative methods (Fig. 3a, b, d and Supplementary Tables 2, 3). For oligosaccharide-binding site prediction (Fig. 3c, d and Supplementary Table 4), DeepGlycanSite showed an average MCC and precision of more than 0.600, and DeepPocket showed an average MCC of 0.410 and an average precision of 0.400. PeSTo showed an average MCC and precision of less than 0.400. The other methods had a small average MCC and precision of less than 0.200, indicating inefficacy in oligosaccharide-binding site prediction. In the nucleotide-binding site prediction, DeepGlycanSite also showed substantially larger average MCC and precision compared with the alternative methods (Supplementary Table 5). Collectively, DeepGlycanSite displayed a great performance across monosaccharide-, disaccharide-, oligosaccharide- and nucleotide-binding site prediction, highlighting its generalized applicability. All methods had reduced performance in glycolipid-binding site prediction (Supplementary Table 6), suggesting such glycoconjugate-binding sites may be distinct from the others and hard to detect.

To compare our model with baseline methods, we have trained two traditional machine learning (ML) models, including SVM and eXtreme Gradient Boosting (XGBoost), on our glycan binding dataset. We extracted feature vectors with 1309 dimensions from our training dataset. The labeling of each amino acid residue was consistent with its classification in DeepGlycanSite. To preserve the integrity of protein data, residues from the same protein were allocated to the same batch for training purposes. DeepGlycanSite significantly outperformed the SVM and the XGBoost models, whose average MCC values were less than 0.200 (Supplementary Table 7). Compared with these two ML models, DeepGlycanSite used the graph representations to effectively capture the correlations between protein residues, contributing to the performance gains.

Considering carbohydrate binding to multiple sites of a protein to be functionally important in several biological processes[30,31], we tested all methods for predicting multiple carbohydrate sites on the testing set TM29. Compared with the alternative methods, DeepGlycanSite obtained significantly better performance with an average MCC of 0.688 and precision of 0.755 (Supplementary Fig. 3a, c and Supplementary Table 8). Meanwhile, in the single carbohydrate-binding site prediction on the testing set TS116, DeepGlycanSite also had an average MCC of 0.609 and an average precision of 0.600, significantly larger than those of the other methods (Supplementary Fig. 3b and Supplementary Table 9). All these results demonstrate the capability and robustness of DeepGlycanSite in addressing complex problems in carbohydrate-binding site prediction.

Since the protein is a flexible molecule, its carbohydrate-bound (holo) conformation could be different from the carbohydrate-free (apo) one. We employed the experimental holo structure dataset T59 and the predicted apo structures dataset T59$_{AF2}$ to assess all methods for carbohydrate-binding site prediction. On both testing datasets, DeepGlycanSite showed superior performance, whose average MCC and precision were approximately twice as those achieved by the other methods (Supplementary Tables 10, 11). These findings validate its effectiveness on the carbohydrate-binding site prediction across various protein conformations.

Ablation experiments on the DeepGlycanSite model were carried out to interpret the network architecture. The elimination of the vector-scalar interactions in the updating units remarkably decreased the performance of the DeepGlycanSite (Supplementary Table 12). Meanwhile, deleting the evolutionary features also led to inferior performance (Supplementary Table 12). In brief, the vector-scalar interactions and evolutionary features are indispensable in DeepGlycanSite.

## Specific binding site prediction for a query carbohydrate

When a given protein binds to multiple different carbohydrates at different sites (Fig. 4a), a reliable prediction method should be able to identify the specific binding site for a query carbohydrate. We build a network DeepGlycanSite$_{+Ligand}$ to process protein structure and the two-dimensional chemical structure of the query carbohydrate with extra modules dealing with ligand parts, including ligand feature extraction and LigandNet (Fig. 4b). The ligand feature extraction module extracts atom and bond features, and global molecular features generated with Uni-Mol[37]. The atom and bond features are fed into LigandNet to produce an output ligand graph, merged with global molecular features into a ligand vector. ReceptorNet takes the ligand vector to produce an output graph. The transformer amalgamates these output graphs to determine the carbohydrate-binding probability of each residue.

We evaluated DeepGlycanSite$_{+Ligand}$ on the testing set TM175, in which each protein binds to multiple different carbohydrates. Because the blind docking strategy has been widely employed to predict ligand-binding sites using both protein and ligand information[38], we compared our method with four state-of-the-art molecular docking

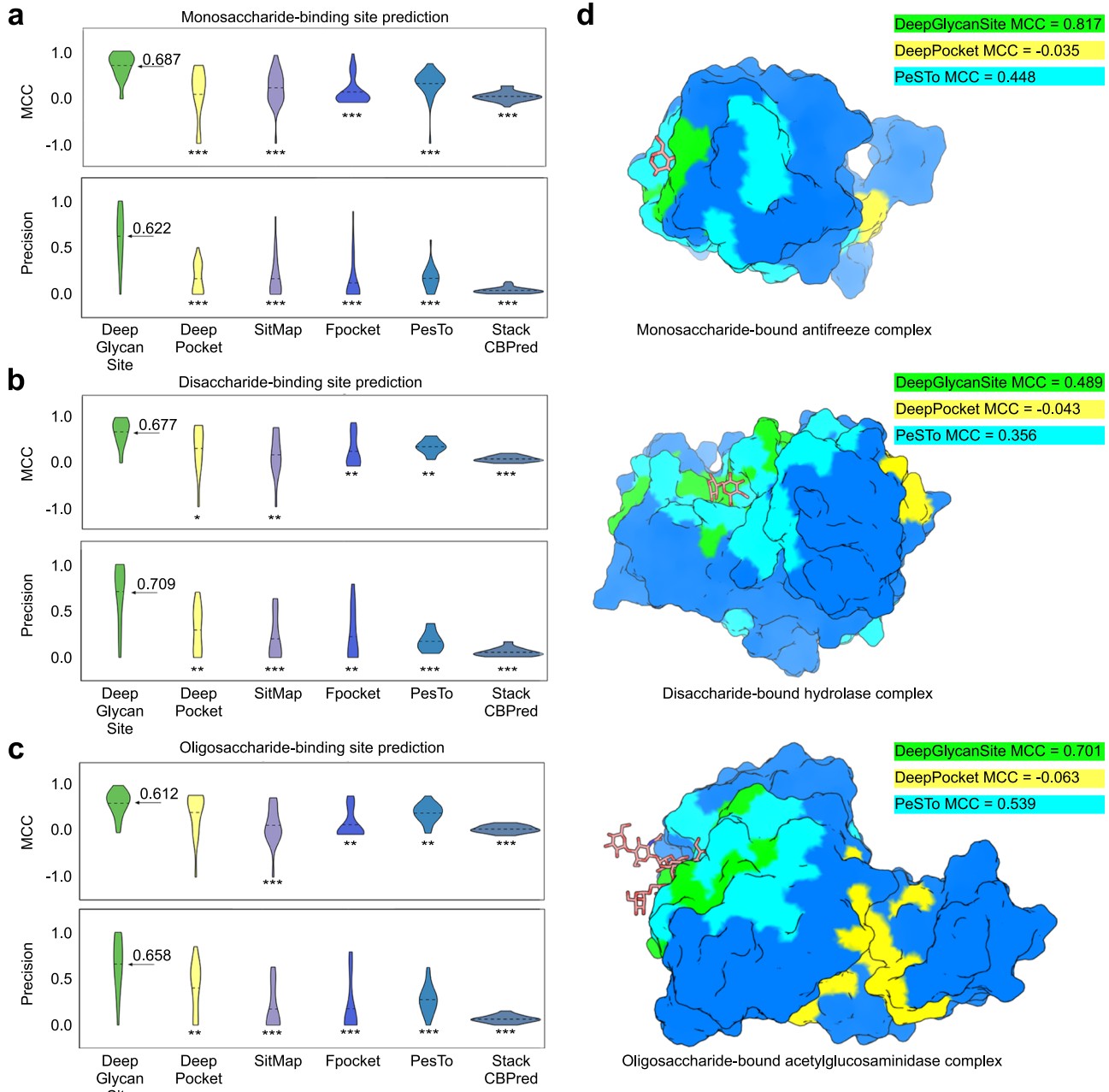

**Fig. 3 | Comparison of model performance in predicting different saccharide-binding sites. a–c** Average Matthews correlation coefficient (MCC) and precision of all methods in predicting mono- (**a**), di- (**b**) and oligosaccharide-binding sites (**c**). The average values are indicated as dashed lines (*n* = 32, 13 and 20). Two-tailed Mann-Whitney U test is used to determine statistical difference. * indicates *P* is less than 0.05, ** indicates *P* is less than 0.01, *** indicates *P* is less than 0.001. From left to right and top to bottom, the *P* values of the significantly different groups are 2.0E-7, 3.2E-7, 6.5E-8, 2.3E-7, 4.9E-10, 2.4E-8, 8.4E-8, 1.7E-8, 1.7E-8, 1.9E-9, 4.0E-2,

2.7E-3, 5.6E-3, 2.3E-3, 2.2E-4, 3.1E-3, 6.0E-4, 1.1E-3, 5.8E-4, 2.7E-4, 1.8E-4, 9.3E-5, 4.7E-3, 1.4E-6, 5.7E-3, 1.5E-5, 5.5E-5, 5.2E-5 and 1.1E-6. **d** Saccharide-binding site prediction of DeepGlycanSite, DeepPocket and PeSTo for three representatives (PDB codes: 6X7X, 7TOH and 7NWF). DeepGlycanSite (green), DeepPocket (yellow) and PeSTo (cyan) predicted binding sites were mapped on the given protein structures. Saccharide molecules are displayed as sticks to indicate the true binding sites. Source data are provided as a Source Data file.

methods (GlycoTorch Vina[39], AutoDock Vina[40], DiffDock[41] and EquiBind[42]) that are feasible for binding site detection. For a molecular docking method, we selected the top-one ranked docking result, and took the carbohydrate-binding residues in this result as the predicted carbohydrate-binding site. As shown in Supplementary Table 13, DeepGlycanSite$_{+Ligand}$ significantly outperformed the alternative methods regarding all metrics (with an average MCC of 0.538, precision of 0.504 and balanced accuracy of 0.806). The robustness of

DeepGlycanSite$_{+Ligand}$ was further validated in the CV experiment (Supplementary Table 14).

To further estimate the performance in distinguishing a specific binding site of a query carbohydrate from the other sites, we categorized binding sites into different classes (Fig. 4c and Supplementary Fig. 4). DeepGlycanSite$_{+Ligand}$ could distinguish the specific binding site of the query carbohydrate belonging to various classes, while the other methods showed inefficacy in distinguishing mono-, di-, or

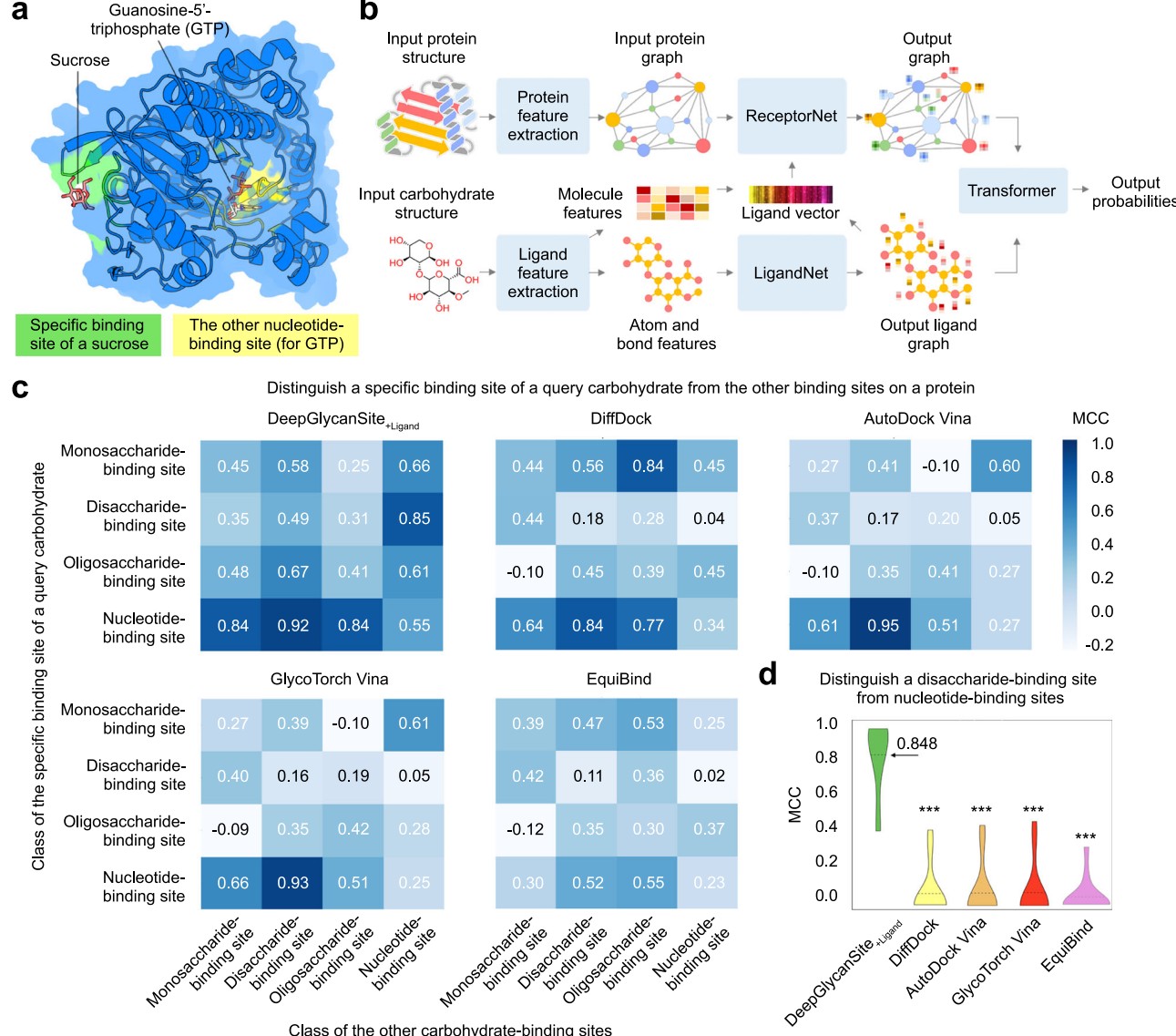

**Fig. 4 | Specific binding site prediction for a query carbohydrate. a** A representative protein binding to two different carbohydrates. Protein is shown in cartoon and surface depict. A disaccharide (sucrose) and a nucleotide (guanosine-5′-triphosphate, GTP) are displayed as sticks to indicate the specific binding site (green) and the other binding site for GTP (yellow), respectively. **b** Model architecture of DeepGlycanSite+Ligand taking the chemical structure of a given carbohydrate and the protein structure for binding site prediction. **c** Heatmaps showing the average Matthews correlation coefficient (MCC) of different methods in predicting a specific binding site of a query carbohydrate when the other binding site exists on the same protein. The carbohydrate binding sites are categorized into four classes to estimate the capability of a method in distinguishing the specific binding site of a query carbohydrate from the other sites. **d** Average MCC of all methods in predicting the specific binding site of a query disaccharide when at least one nucleotide-binding site exists on the same protein. The average values are indicated as solid lines ($n = 12$). Two-tailed Mann-Whitney U test is used to determine statistical difference. *** indicates $P$ is less than 0.001. From left to right, the $P$ values of the significantly different groups are 4.3E-5, 5.7E-5, 5.7E-5, and 3.5E-5. Source data are provided as a Source Data file.

oligosaccharide-binding sites with small average MCC and precision less than 0.200 (Fig. 4c and Supplementary Fig. 4a).

Notably, DeepGlycanSite+Ligand outperformed the alternative methods in distinguishing the specific binding site of a query disaccharide from the nucleotide-binding sites (Fig. 4d and Supplementary Fig. 4b). In a representative case of the RNA 2′,3′-cyclic phosphate and 5′-OH ligase (RtcB), only DeepGlycanSite+Ligand successfully identified the specific binding site of sucrose, while all the other methods predicted that the disaccharide binds to the nucleotide-binding site (Supplementary Fig. 5). These results suggest the remarkable ability of DeepGlycanSite+Ligand in identifying specific carbohydrate-binding sites. We also conducted ablation experiments of the DeepGlycanSite+Ligand, which indicates the ligand

information, especially the ligand vector, as a key component for the model (Supplementary Table 15).

### Experimental validation of DeepGlycanSite prediction
We used DeepGlycanSite+Ligand to identify the specific carbohydrate-binding site on a functionally important GPCR, i.e., P2Y purinoceptor 14 (P2Y14), which regulates immune responses and associates with asthma, kidney injury and lung inflammation[43,44]. In the calcium mobilization assay, we found that the guanosine 5′-diphosphate-fucose (GDP-Fuc) activates human P2Y14 with the half-maximal effective concentration ($EC_{50}$) of $0.49 \pm 0.04\ \mu M$. As an essential sugar nucleotide in mammals, the GDP-Fuc is critically involved in tumor growth and metastasis in various cancers[45–47]. The GDP-Fuc-induced

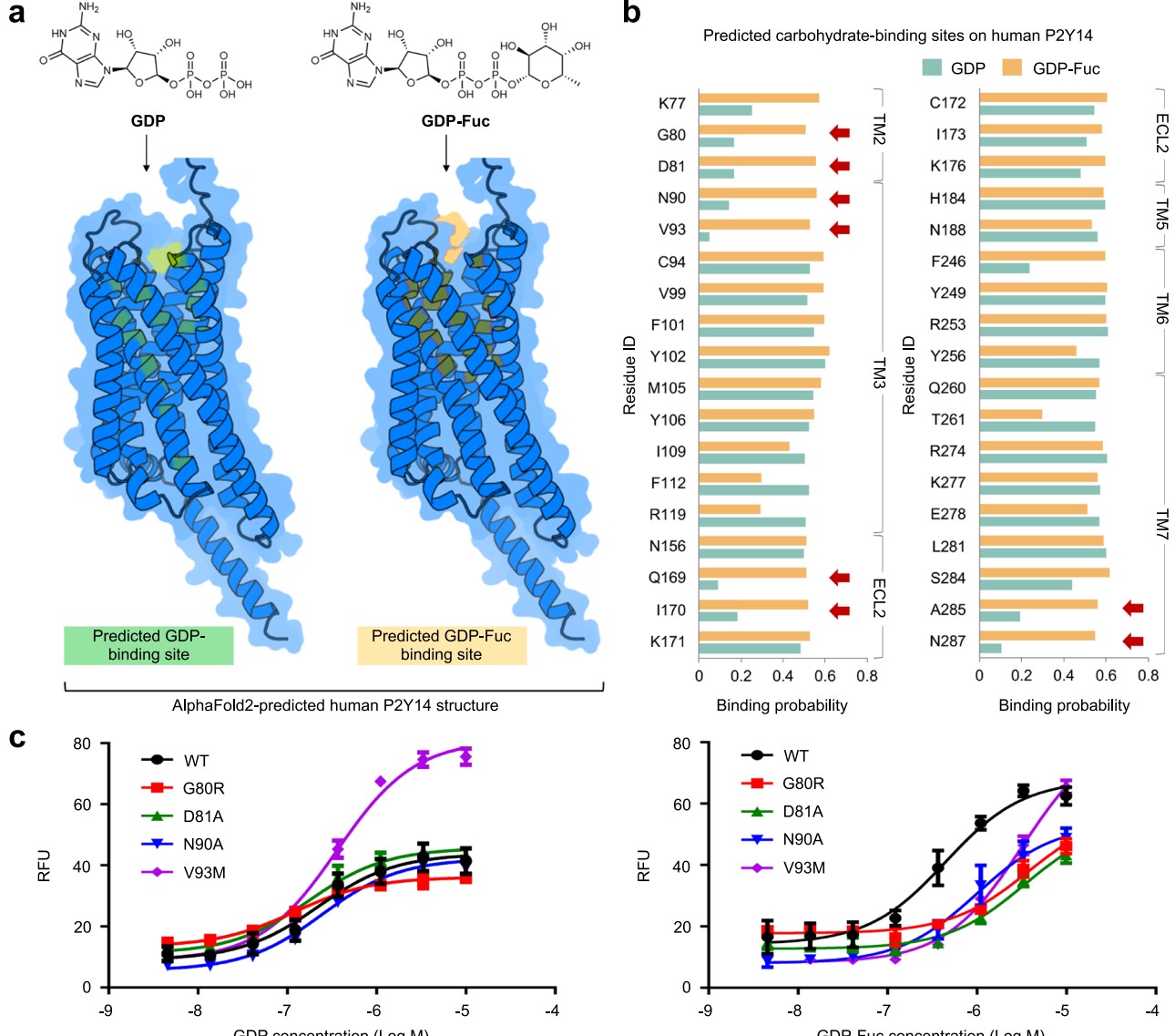

**Fig. 5 | Experiment validation of DeepGlycanSite+Ligand. a** Predicted binding sites of guanosine-5′-diphosphate (GDP) and guanosine-5′-diphosphate fucose (GDP-Fuc) by DeepGlycanSite+Ligand on the AlphaFold2-predicted human P2Y14 structure. **b** Binding probabilities of predicted carbohydrate-binding sites for GDP and GDP-Fuc, respectively. The predicted fucose-binding residues are labeled with red arrows. **c** Calcium mobilization concentration-response curves for GDP or GDP-Fuc in HEK293 expressing P2Y14 wild-type (WT) and mutants. Data were presented with a minimum of three independent biological replicates. The error bar indicates standard error. Source data are provided as a Source Data file.

activation of P2Y14 has not been reported before. Hence, how GDP-Fuc acts on this receptor is unknown. GDP also activates P2Y14[48]. We assumed that both GDP-Fuc and GDP directly bind to and act on the P2Y14. By comparing the DeepGlycanSite+Ligand-predicted binding sites of GDP-Fuc and GDP, we tried to identify the specific binding site for the fucose moiety. Because the experimental structures of P2Y14 are unavailable, we used the AlphaFold2-predicted structure model[49] for analysis (Fig. 5a).

The predicted GDP- and GDP-Fuc-binding sites encompassed five transmembrane helices (TMs 2, 3, 5, 6 and 7) and an extracellular loop 2 (ECL2) (Fig. 5b). We identified eight possible fucose-moiety-binding residues (G80, D81, N90, V93, Q169, I170, A285 and N287), whose GDP-binding probability is less than 0.2 while GDP-Fuc-binding probability is more than 0.5 (Fig. 5b). To validate the prediction, we designed single-point mutations for these eight predicted binding residues and four surrounding residues (L79, V91, F92 and A286). Among them, single-point mutations of four predicted binding residues (G80, D81,

N90 and V93) significantly reduced the GDP-Fuc-induced receptor activities (Fig. 5c and Supplementary Table 16). The potencies of GDP-Fuc reduced by five- to ten-fold on three mutants (G80R, D81A and V93M), compared with that in the wild-type (WT) group (Fig. 5c and Supplementary Table 16). In contrast, these three mutations did not show a great effect on the reduction of GDP-induced receptor responses (Fig. 5c and Supplementary Table 17). These results validate that G80, D81 and V93 are key residues for fucose moiety-recognition of P2Y14. Substitute of N287 by alanine diminished GDP-Fuc and GDP-induced responses (Supplementary Fig. 6). However, other mutations did not exhibit great impact on receptor activation by GDP-Fuc or GDP (Supplementary Fig. 6 and Supplementary Tables 16, 17). These results not only validate our assumption that GDP-Fuc directly binds to and acts on P2Y14 but also identify the fucose moiety-recognition residues.

To enhance our understanding of the fucose-moiety recognition of P2Y14, we tried to construct a reliable GDP-Fuc-bound P2Y14 complex model. We placed the GDP-Fuc in the center of the

DeepGlycanSite$_{+Ligand}$-predicted carbohydrate-binding site of the AlphaFold2-predicted P2Y14 structure, and then performed molecular dynamics (MD) simulation to refine the complex model. In simulations, the hydroxyl group of the fucose moiety could interact with D81 and N90 via water-medicated hydrogen bonds, and the methyl group of fucose could form hydrophobic interactions with V93 (Supplementary Fig. 7). We also performed molecular docking of GDP-Fuc to the AlphaFold2-predicted P2Y14 structure model to construct the complex model. However, the top-one ranked docking models generated by AutoDock Vina, GlycoTorch Vina and EquiBind had the fucose moiety far from D81, N90 and V93 (Supplementary Fig. 8), inconsistent with our experimental results. DiffDock failed to produce any docking results.

## Discussion

In this study, we show that DeepGlycanSite is a highly accurate approach to predicting the carbohydrate-binding sites on proteins, outperforming alternative methods. We compared DeepGlycanSite with StackCBPred, which is a tailored method of sequence-based carbohydrate-binding site prediction. The average MCC of DeepGlycanSite (0.625) is more than thirty-fold that of StackCBPred (0.018) (Table 1). The other previous sequence-based carbohydrate-binding site predictors SBRP[15] and SPRING-CBH[16] reported small average MCC values of 0.200 and 0.270, respectively. All these three methods used the SVM algorithms with protein sequence features, including the position-specific scoring matrix, and several predicted properties of amino acids[14–16]. Tsai et al. developed a machine learning algorithm that used the three-dimensional probability density maps to describe the carbohydrate-interacting atoms around the protein surface[50], and reported an average MCC of 0.45 on an independent test set of 108 proteins[50]. Zhao et al. developed a template-based predictor of carbohydrate-binding proteins SPOT-Struc[51] and achieved an average MCC of 0.51 on a test set of 59 carbohydrate-binding proteins[51]. Because SBRP, SPRING-CBH, Tsai's method and SPOT-Struc are no longer available, we cannot directly compare them with DeepGlycanSite. Nevertheless, compared with these previous approaches using protein sequence or structure information, DeepGlycanSite integrates the sequence-based evolutionary features with the structure-based geometric features, which may provide a more comprehensive description of the carbohydrate-binding sites and promote the prediction performance. As a modern deep learning method, DeepPocket uses a voxel-based convolutional neural network to predict ligand-binding pockets. Different from DeepPocket, DeepGlycanSite employs an EGNN, handling the sparse connections among nodes of the network. Both DeepGlycanSite and PeSTo are EGNN models converting geometric information into vectors and scalars for feature representation. Nevertheless, DeepGlycanSite leverages a more sophisticated message-passing architecture, which makes full use of vectors and scalars by updating both node and edge features (Supplementary Fig. 1a). But PeSTo only updates node features after message passing. In addition to the geometric information, DeepGlycanSite considers the evolutionary features of the protein. Both enhanced EGNN and the evolutionary features may contribute to the superior performance of DeepGlycanSite, compared with the previous methods.

DeepGlycanSite is a robust binding site predictor, showing consistently good performance across different carbohydrate-binding site classes. In predicting monosaccharide- or disaccharide-binding sites, DeepGlycanSite had a significantly larger average MCC and precision compared with the other ligand-binding predictors (DeepPocket, SiteMap, Fpocket and PeSTo) (Fig. 3 and Supplementary Tables 2, 3). In a typical process of ligand-binding pocket (or site) prediction, hydrophilic or small putative binding pockets (or sites) may be removed in pruning of uninteresting pockets (or sites)[13]. Small and essentially polar pockets (or sites) are considered to be less likely to interact with

small-molecule ligands and therefore dropped from the protein surface[13]. The binding sites of simple carbohydrates, i.e., monosaccharides and disaccharides, are mostly small and polar, which may be excluded by the traditional ligand-binding site predictors. This may explain the reduced performance of ligand-binding site predictors in detecting monosaccharide- or disaccharide-binding sites.

DeepGlycanSite presents an outstanding performance in predicting multiple carbohydrate-binding sites on proteins (Fig. 4 and Supplementary Tables 8, 13). Unlike small-molecule ligand binding sites, carbohydrate-binding sites can be found in a multitude of protein folds[52]. Because carbohydrate-protein interactions can be low affinity, carbohydrates and carbohydrate-binding proteins could present in multiple-to-one relationships to enhance the affinity through avidity. The carbohydrate-binding to multiple sites within the target protein exerts a regulatory role in several biological processes[3,53–56]. For example, carbohydrates bind to multiple sites of sialic acid-binding immunoglobulin-type lectin 2 (sigelc2) to carry cargo into B cells to mediate immune homeostasis[3,53,54]. Multivalent interactions between lectins and carbohydrates result in an overall increase in binding affinity, contributing to the regulation of innate and adaptive immunity[55,56]. Multivalent glycoconjugates have been designed as chemical tools and drug candidates to influence carbohydrate-lectin interactions[57]. As a prediction method capable of detecting multiple carbohydrate-binding sites, DeepGlycanSite may help identify different carbohydrate-binding domains on a protein and provide crucial insights into the carbohydrate-regulated synergistic (or multivalent) mechanisms.

Given the AlphaFold2-predicted protein structure and the carbohydrate chemical structure, our method successfully detected the specific binding site of GDP-Fuc on human P2Y14. Some recent studies have shown that the side chain quality modeled by AlphaFold2 is not satisfactory, and therefore the docking test based on AlphaFold2-predicted protein structure showed weak enrichment performance[58,59]. Consistently, employing AlphaFold2-predicted structure, AutoDock Vina, GlycoTorch Vina and EquiBind can hardly generate GDP-Fuc-P2Y14 docking models consistent with our mutagenesis data (Supplementary Fig. 8). Moreover, DiffDock failed to produce results. Compared with these molecular docking methods that rely on the accuracy of local structure details or sidechain conformations of the receptor, our method is less sensitive to the protein structure accuracy (Supplementary Table 11) and could provide insight into the carbohydrate-protein interaction using predicted protein structures.

In conclusion, the validation of the DeepGlycanSite predictions on the independent test sets and in the in vitro case study offers us confidence that DeepGlycanSite is an effective tool in carbohydrate-binding site prediction. Researchers could employ the DeepGlycanSite to predict carbohydrate-binding pockets on the target proteins, which can be either experimentally determined or predicted structures, to facilitate the investigation of carbohydrate-protein interactions. Carbohydrates are critical mediators of biological function. Their remarkably diverse structures and varied activities present exciting opportunities for understanding many areas of biology. We hope the DeepGlycanSite will not only help decipher the biological functions of carbohydrates and carbohydrate-binding proteins but also provide a powerful tool for the development of carbohydrate drugs.

## Methods
### Datasets
Our criterion for labeling the data is the massive-atom distance: only residues within 4-Å distance from carbohydrates are labeled as carbohydrate-binding sites. We obtained X-ray and electron microscopy structures of carbohydrate-protein complexes from the PDB database with a maximum release date of Jan 1, 2023, and a resolution better than 4 Å to construct datasets. Glycosylated carbohydrates were

removed from the complex structures. We employed the complexes from 2020 or earlier for training or validation. The dataset was sequentially split into an 80% training set and a 20% validation set. The complexes released after 2021 with a resolution better than 3 Å were used to construct independent testing sets. To reduce possible bias toward some popular proteins and consider more protein-carbohydrate interactions, for the training set, we excluded any instance of the same site binding to the same carbohydrate. For the testing set, any protein with more than 95% sequence identity to those of the training (or validation) set was excluded. We further controlled protein sequence identity of 30% within the testing set to construct a nonredundant independent dataset T145. The T145 involves 145 carbohydrate-protein complexes. In the T145 testing set, a carbohydrate (or several different carbohydrates) binds to multiple pockets of a protein in 29 complexes (TM29), while a carbohydrate binds to only one pocket of a protein in 116 complexes (TS116). In addition, we excluded proteins of T145 with more than 25% homology from the training (or validation) set to construct an independent testing set T59. T59 involved 59 carbohydrate-binding proteins. To obtain various structures of the same protein, we employed AlphaFold2[32] and AlphaFold2 Multimer[33] to predict protein structures based on the protein sequences of T59. The top five ranked conformation models for each protein were selected to construct an independent testing set T59$_{AF2}$, consisting of 59 unique proteins and 295 apo structure models. We also constructed a testing set of 175 unique complex samples, termed TM175, in which a protein binds to distinct carbohydrates at different sites. We compared the carbohydrate-binding sites of TM175 with those of the training and validation sets using Foldseek[60], a tool for structural similarity comparison. There are no similar sites between the TM175 set and the training (or validation) set with more than 80% identity. The training, validation and independent test sets are available for download at https://github.com/xichengeva/DeepGlycanSite/tree/main/datasets.

## Features for graph representation

We utilized undirected graphs [G = (V, E)] to represent proteins and carbohydrates. For a protein, a node was assigned for each residue, and an edge connecting two neighboring residues within an 8-Å massive-atomic distance threshold in the given protein structure. The position of each node was defined by the center of mass of each residue. Node features included residue type, embedded evolutionary information, and intra-residual geometric features. We used the ESM-2 model (esm2_t33_650M_UR50D)[24] to generate 1280-dimensional embedding evolutionary information based on the amino acid sequence of the given protein. Different chains in one protein were divided before the generation of ESM information. The intra-residual geometric features consisted of distances and dihedrals. The position of the residue served as the node coordinates for vector calculations. Edge features included residue connectivity and inter-residual geometric features that define the relative distance and orientations of two neighboring residues, i and j[61]. Supplementary Tables 18, 19 summarized the input features for the protein and provided an in-detail description of how they were calculated.

For carbohydrates in DeepGlycanSite$_{+Ligand}$, nodes and edges represent the atoms and bonds, respectively. A 512-dimensional molecule feature was introduced for global featurization. Rdkit[62] processed a query carbohydrate before featurization. Node features included atom symbol, degree, hybridization type, formal charge, number of radical electrons, aromaticity, total number of hydrogens binding on it, and chiral property. Edge features incorporated bond type, conjugation, ring inclusion and stereo configuration. Supplementary Tables 20, 21 summarized the input features for the carbohydrates and described how they were calculated. The molecule feature was calculated with the SMILES of the carbohydrate, using Rdkit and Uni-mol (mol_pre_all_h_220816.pt[37]).

## Model architecture

Inspired by the Vector-Scalar Interactive Graph Neural Network (ViSNet), we constructed the ReceptorNet as a geometry-based equivariant graph neural network to decipher residue-level representations for carbohydrate-binding site prediction. Below, we provide an elucidation of how the network facilitates information exchange among neighboring residues. Following the residue feature extraction and embedding, node features $f_n$ were initially projected into hidden dimensions as Eq. (1).

$$h_n = W_n(f_n) \qquad (1)$$

$W_n$ denote linear weights for projecting features toward hidden dimensions $h_n$ without additional bias. Subsequently, the distance between two interconnected nodes was normalized by an exponentially modified Gaussian radial distribution function $g(\vec{r}_{ij})$ (Supplementary Methods). Then edge hidden dimensions were calculated via Eq. (2) according to the sum of edge features ($f_e$) and radial distribution function projection.

$$h_e = W_e(f_e) + W_r\left(g\left(\vec{r}_{ij}\right)\right) \qquad (2)$$

$W_e$ and $W_r$ denote linear weights for projecting edge features and the output of radial distribution function towards edge hidden dimensions $h_e$, respectively. Neighborhood embedding was then applied to mix node features. In each node and edge, a vector was initialized to show direction.

As depicted in Supplementary Fig. 1a, these node features, edge features, node vectors and edge vectors were passed through attention layers, yielding the delta value for updating. The process includes two main modules: Scalar2Vec and Vec2Scalar. The Scalar2Vec module is designed to update vectors using scalar features, while the Vec2Scalar module performs the reverse process, updating scalar features based on vectors. The Scalar2Vec module is described in Eq. (3).

$$\Delta\vec{v}_i^{l+1} = \vec{m}_{ij}^l + W_{vm}^l m_i^l \odot W_v^l \vec{v}_i^l \qquad (3)$$

Where $l$ is the index of the attention layer, $m_{ij}^l$ and $\vec{m}_{ij}^l$ mean intermediate scalar and vector for updating (Supplementary Methods), $\odot$ is the Hadamard product, and $\Delta\vec{v}_i^{l+1}$ represents the delta value for updating vector embedding.

Vec2Scalar module updates the node and edge embedding using the geometric information in vectors. For node updating, the delta value $\Delta n_i^{l+1}$ is calculated according to Eq. (4)

$$\Delta n_i^{l+1} = \left\langle W_r^l \vec{v}_i^l, W_s^l \vec{v}_i^l \right\rangle \odot W_a^l m_i^l + W_r^l m_i^l \qquad (4)$$

Where angle brackets represent the inner product for different weight projections on vectors. For edge updating, the delta value $\Delta e_{ij}^{l+1}$ is calculated according to Eq. (5)

$$\Delta e_{ij}^{l+1} = \left\langle Rej_{\vec{r}_{ij}}\left(W_{Rt}^l \vec{v}_i^l\right), Rej_{\vec{r}_{ji}}\left(W_{Rs}^l \vec{v}_j^l\right) \right\rangle \odot Dense_D^l e_{ij}^l \qquad (5)$$

$Rej_{\vec{r}_{ij}}$ represents rejection on the vector and $Dense$ function refers to one learnable weight matrix with a sigmoid linear unit activation function. Each delta value is cumulatively added to the previous value for updating. Next, node features and node vectors are input to a gated equivariant block[63], yielding the final outputs as node features.

Subsequently, these node features served as inputs for a transformer decoder structure without mask and positional encoding, with encoder input set to zeros. Such a structure was designed for further integration of residue features, especially for the nodes that are not connected. Lastly, node features went through a linear layer to reduce

the dimension to one, and a sigmoid function was applied to generate the final output, representing the probability of carbohydrate binding.

For DeepGlycanSite$_{+Ligand}$, the carbohydrate features of nodes and edges were initially projected into hidden dimensions using linear layers. They underwent an update process in LigandNet, including the MetaConv and ResBlock layers, which are described in the Supplementary Methods and Supplementary Fig. 1b. A 'set2set' operation was applied to ultimately extract the graph-level features of the carbohydrate as a vector[64]. Subsequently, these vectors and pretraining molecule features were merged into a so-called ligand vector, as illustrated in Eq. (6).

$$Vec_{Lig} = MLP_{ds3}([set2set_{out}, Vec_{Pretrain}]) \qquad (6)$$

$MLP_{ds3}$ down-samples the dimensions to one-third of the input, which adjusts the hidden dimensions. Then, the ligand vector was replicated the number of times equal to the receptor node count. Such vector was integrated into both the protein graph node features and the delta node features within the attention layers as demonstrated in Eq. (7) and (8), which were modified from Eqs. (1) and (4).

$$h_n = [W_n(f_n), Vec_{Lig}] \qquad (7)$$

$$\Delta n_i^{l+1} = \left\langle W_r^l \vec{v}_i^l, W_s^l \vec{v}_i^l \right\rangle \odot W_a^l m_i^l + W_r^l m_i^l + Vec_{Lig} \qquad (8)$$

These updated node features for carbohydrates also served as the encoder input for the final transformer, facilitating the fusion of carbohydrate information with receptor data in our neural network. Finally, per receptor node also went through a linear layer and a sigmoid function to generate its probability of specific carbohydrate binding.

### Training
We used WeightedFocalLoss[65] (Eqs. (9–10)) to measure the loss for the residues being evaluated as carbohydrate-binding sites.

$$p_y = -[y * \log(p) + (1 - y) * \log(1 - p)] \qquad (9)$$

$$Loss(p, y) = -\alpha * (1 - p_y)^\gamma * \log(p_y) \qquad (10)$$

Where $p$ is the predicted probability of the residue, $y$ is the target label (0 or 1), $\alpha$ is a weighting factor for class $y$, and $\gamma$ is a focusing parameter that controls the contribution of easy and hard examples.

Hyperparameters were optimized through Bayesian optimization[66]. In a five-fold cross-validation, we identified the top-performing model on the validation set as the final model according to the average MCC. The test results of cross-validations are shown in Supplementary Tables 1, 14. The hyperparameters of DeepGlycanSite and DeepGlycanSite$_{+Ligand}$ are shown in Supplementary Table 22. The AdamW optimizer with the ReduceLROnPlateau strategy was implemented. In the training of DeepGlycanSite, both the radial basis function and vector normalization parameters are trainable. The model was trained using the DDP strategy on four 80 GB A100 GPUs for ~30 hours. In the training of DeepGlycanSite$_{+Ligand}$, both radial basis function and vector normalization parameters are not trainable. The model was trained on a single 80 GB A100 GPU for ~40 hours.

### Molecule docking
A blind redocking was performed to assess the performance of a docking method in the carbohydrate-binding site prediction. For AutoDock Vina[40] or GlycoTorch Vina[39], a cubic grid box, centered on the protein and extending 5 Å from the surface was defined for docking. For Diffdock[41] and EquiBind[42], we used its default setting for blind docking. Further details are provided in the Supplementary Methods.

### Molecular dynamics simulations
Initially, a ligand was placed in the center of the predicted binding site and minimized using Schrödinger's Maestro to avoid conflict. To build a simulation system, we placed the complex model into a 1-palmitoyl-2-oleoyl-sn-glycero-3-phosphocholine (POPC) lipid bilayer. The lipid-embedded complex model was solvated in a periodic boundary condition box (60 Å × 60 Å × 135 Å) filled with TIP3P water molecules[67] and 0.15 M NaCl using CHARMM-GUI[68]. Each system was replicated to perform three independent simulations. Based on the CHARMM36m all-atom force field[69] for protein and CHARMM general force field[70] for ligand, molecular dynamics simulations were conducted using Amber20 pmemd.cuda[71]. The system setups are shown in Supplementary Table 23. After the minimization process of 5000 steepest descent cycles with a constraint on backbone atom, sidechain atom, and lipid coordinates and a constraint on dihedrals, the constraints were generally decreased in the separated 6 steps of the equilibration process provided by CHARMM-GUI. 500 ns production run was then carried out for each simulation.

All productions were carried out in the NPT ensemble at a temperature of 303.15 K and a pressure of 1 atm. Temperature and pressure were controlled using the Nose-Hoover thermostat[72] and the Parrinello–Rahman barostat[73], respectively. Equations of motion were integrated with a 2-fs time step as the SHAKE algorithm was used to constrain bond length[74]. Long-range electrostatic interactions were addressed using the Particle Mesh Ewald method[75]. Short-range electrostatic and van der Waals interactions were treated with a 12 Å cutoff, which was gradually switched off between 12 Å and 10 Å. These all-atom simulation models could provide important information for the protein-ligand interactions[76–79]. The final 200 ns trajectory of each simulation was used for the extraction of representative structure.

Amber20 CPPTRAJ "cluster" program was applied to the extraction. Firstly, per 200 ps we input 1 structure, then 1000 snapshots were obtained. Then, a hierarchical agglomerative (bottom-up) approach was applied based on the average distance between members of two clusters. The distance between snapshots was calculated as the best-fit coordinate RMSD using all heavy atoms. The clustering stops when 1 cluster remains and the structure whose RMSD is smallest to the other structures was picked as the representative one.

### Evaluation metrics
There are mainly four metrics used to evaluate carbohydrate-binding site detection algorithms. The metrics are the Matthews correlation coefficient (MCC), precision, and balanced accuracy. For a given protein, the carbohydrate-binding residues are positive, while the others are negative. Correctly predicted carbohydrate-binding residues are true positives (TP). Correctly predicted non-carbohydrate-binding residues are true negatives (TN). Incorrectly predicted carbohydrate-binding residues are false positives (FP). Incorrectly predicted non-carbohydrate-binding residues are false negatives (FN). Further details are provided in Supplementary Methods.

To evaluate the performance of a ligand-binding site (or pocket) detection method, i.e., DeepPocket, PeSTo, Fpocket, or SiteMap, we took the top-N ranked unique predicted site (or pocket) for estimation, where "N" is the number of the true carbohydrate-binding sites (or pockets) for the given protein. To evaluate the performance of molecular docking methods, i.e., Autodock Vina, GlycoTorch Vina, DiffDock or EquiBind, we took the top-one ranked docking model for estimation. In such docking models, residues within 4-Å massive-atomic distance from the query carbohydrate defined the predicted carbohydrate-binding site.

## Chemical materials

GDP was purchased from Sigma-Aldrich (G7127). GDP-Fuc was synthesized according to the reported work[80].

## Cell culture and transient transfections

HEK293 cells were cultured in Dulbecco's modified Eagle's medium (DMEM) with 10% fetal bovine serum (FBS). All cells were maintained at 37 °C in humidified incubators with 5% $CO_2$ and 95% air. Human P2Y14 receptor and G protein α-subunit ($G\alpha_{qi5}$) were transiently co-transfected into HEK293 cells using PolyJet in vitro DNA transfection reagent (SignaGen) according to manufacturer's instructions. Thus, a mixture of 1 μg of receptor DNA and 1 μg of $G\alpha_{qi5}$ DNA was used to transfect into the 6-well plate cells at 90% confluency. HEK293 cells transiently expressing the P2Y14 receptor were subsequently used for the intracellular $Ca^{2+}$ assays at 48-hour post-transfection.

## Intracellular calcium mobilization

Intracellular calcium assays were carried out as follows. HEK293 cells were seeded (80000 cells/well) into a Matrigel-coated 96-well plate for 24 hours before assay. The cells were incubated with 2 μM Fluo-4 AM (Invitrogen) diluted in HBSS solution (meilunbio) at 37 °C for 50 minutes. After dye loading, the cells were treated with the compounds of interest. Then, calcium response (relative fluorescence unit, RFU) was measured using Flexstation 3 (Molecular Device) with fluorescence excitation made at 485 nm and emission at 525 nm.

## Statistics

Statistical analyzes were performed using GraphPad Prism 6 (GraphPad Software). EC50 values for compounds were obtained from concentration-response curves by nonlinear regression analysis. The comparison of the two constructs was analyzed by unpaired t-test to determine statistical differences. All statistical data are given as mean ± standard error of mean (SEM) of at least three independent biological replicates.

## Reporting summary

Further information on research design is available in the Nature Portfolio Reporting Summary linked to this article.

## Data availability

All input data used in this study are freely available from PDB (https://www.rcsb.org), AlphaFold (https://alphafold.com) and PubChem (https://pubchem.ncbi.nlm.nih.gov). The training, validation and independent test sets are available for download at https://github.com/xichengeva/DeepGlycanSite/tree/main/datasets. Source data are provided in this paper. The initial coordinate, simulation input files, and coordinate files of the final output were provided at https://doi.org/10.5281/zenodo.11208156. Source data are provided in this paper.

## Code availability

The source code, final model and associated data preparation scripts are available at GitHub (https://github.com/xichengeva/DeepGlycanSite). To cite our code, please refer to https://doi.org/10.5281/zenodo.11201294[81].

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

## Acknowledgements

We would like to express our deepest gratitude to Prof. Hualiang Jiang, who provided invaluable guidance and support throughout this work. Although Prof. Jiang is no longer with us, his wisdom and mentorship continue to inspire us. We thank Prof. H. Eric Xu for the valuable discussions. This work is partially supported by Shanghai Municipal Science and Technology Major Project (X.C. and L.W.), National Key Research and Development Program of China 2021YFA1301900 (X.C.), Fund of Youth Innovation Promotion Association 2022077 (X.C.) and the Lingang Laboratory LG202102-01-01 (X.C.).

## Author contributions

The overall concepts in the paper were developed, and supervision was carried out by X.C., D.W., and L.W. X.H. who developed the algorithm. L.Z. performed experimental validation. Y.T. synthesized chemicals. R.L., Q.C., Z.G., M.Z., Y.W., S.L., H.J., and Y.J. carried out the analyses and prepared related figures and tables. All authors contributed to writing the manuscript.

## Competing interests

The authors declare no competing interests.
