## [Peer Review File · Nature Communications]

Reviewers' Comments:

Reviewer #1:

Remarks to the Author:

This manuscript described a new deep learning method to predict carbohydrate-binding residues. They integrated geometric and evolutionary features into GCN for the prediction, a routine technique in protein prediction studies. The method was compared with other methods and validated on a GPCR. Generally, the glycan binding site prediction isn't a new problem and a few prediction methods were proposed. The model doesn't show enough novelty in methodology, and the experimental validation has flaws as detailed below.

1. The methodology doesn't show enough novelty. Many recent techniques like equivalent GCN has shown improvements over this employed original GCN architecture. Though the method was shown to outperform other methods like Pesto, these methods themselves were designed for binding site with ligands, not for glycans. It's interesting to re-train these modern models on this glycan binding dataset or re-implement traditional ML methods to see the performance. Alternatively they can indicate the method on the well-studied PPI or DNA binding sites prediction. On the other hand, the authors can compare with the recently developed general binding site(PMID: 37054909).
2. For the current dataset, the authors only control sequence identity of 30% (generally should be 25%) between training/test set but not excluding redundant sequences within training/test sets. This may cause bias toward some popular protein family. In addition, a CV on the training set is also needed to indicate the robustness of the method.
3. Current wet experimental didn't validate their model. First only one out of 8 predicted showed significantly decreased binding affinity, indicating the poor prediction binding site. Secondly, they only mutated predicted binding residues but didn't employ other residues as control. The drop in the binding affinity can be caused by direct interaction, and also by the impact on the overall structure. They can select to mutate AA in surrounding residues (like #286) and residues in other regions as a contrast, or they can determine the complex structure.

Reviewer #2:

Remarks to the Author:

In this paper, the researchers introduce DeepGlycanSite, a deep learning model designed to forecast carbohydrate binding sites across diverse proteins. Additionally, they conducted a comparative analysis with existing software for detecting carbohydrate binding sites, demonstrating superior performance using their proposed method. They also applied the method to human P2Y14 and confirmed their results with experimental observations.

Here are my comments:

1. In the introduction, include that a challenge associated with the utilization of carbohydrate-based drugs also lies in their specificity. This limitation arises not only due to a lack of comprehension regarding their binding mechanisms but also because even when such understanding exists, achieving the desired specificity proves to be a challenging task.
2. If I grasp the information correctly, you employed PDB structures in training the model, where the input consists of protein structures from which carbohydrates were extracted. The test set also employed a similar structure type. My concern stems from the potential bias in such a structure, wherein the mere presence of a carbohydrate within it might influence the model to identify the attachment site. This bias arises from the protein structure remaining unaltered compared to the one with the attached carbohydrate. Can you comment on this?
3. Have you attempted to assess your model using various structures of the same protein? Testing the model across diverse structures would serve as a robust validation, especially considering the exclusion of proteins with more than 30% homology from both the training and testing sets. Achieving consistent results across multiple structures would strongly affirm the effectiveness of

your model.

4. Conversely, the AlphaFold model underwent equilibration, and a representative structure was selected following a simulation lasting 300-500 ns. While the binding site prediction was accurate, it's worth noting that this site corresponds to a region of the GPCR protein that other software similarly identifies as a binding site, primarily due to its proximity to the orthosteric binding site.

5. Have you tried with conducting MD simulations on a select set of representative examples and subsequently applied your protocol to assess a few representative structures? This approach would help determine the robustness of your method against conformational changes. Additionally, such testing could mitigate the bias introduced by the presence of carbohydrates in the structure. I would strongly suggest this.

Reviewer #3:

Remarks to the Author:

The authors developed a deep learning model DeepGlycanSite to predict carbohydrate-binding sites. The PDB dataset was used for training and validation after removal of homologous proteins. The performance of DeepGlycanSite is better than the previous methods including DeepPocket and PeSTo which are also based on deep learning methods. Finally, the authors experimentally validated the prediction of DeepGlycanSite on a GPCR P2Y14. Specific comments are listed below:

Major points:

1) In general, subtle differences in carbohydrate binding specificity result from limited amino acid changes in the ligand binding site of a same protein fold. For example, EPN and QPD motifs are presented on a common C-type lectin fold but bind different sugars, Man and Gal. Legume lectins share a common fold but have different binding specificities. The authors omitted homologous proteins for training and validation and I am afraid that such subtle differences in glycan ligands may not be considered in DeepGlycanSite. In other words, proteins of a same protein fold can bind to different sugars. Typical lectin folds are C-type lectins, legume lectins, and I-type lectins etc. This point needs to be validated and discussed.

2) Relatively good performance was seen for predicting nucleotide binding. Did DeepGlycanSite distinguish the sugar nucleotides, such as GDP-Fuc and GDP-Man that share a common nucleotide but have different sugars? For example, how was the prediction of GDP-Man binding to P2Y14?

3) DeepPocket and PeSTo use deep learning methods but DeepGlycanSite showed better performance than these methods. What makes DeepGlycanSite different from these methods?

4) The description of the MD simulation of the P2Y14 complex is very poor and I could not find out how the representative model was selected. The relevant data and criteria must be presented. Otherwise, the model cannot be treated as a reliable model even though experimental results are available.

Minor point:

Line 48, 50: 200 kDa will be 200 Da. 1000 kDa will be 1000 Da.

Point-by-point response

Reviewer #1

This manuscript described a new deep learning method to predict carbohydrate-binding residues. They integrated geometric and evolutionary features into GCN for the prediction, a routine technique in protein prediction studies. The method was compared with other methods and validated on a GPCR. Generally, the glycan binding site prediction isn't a new problem and a few prediction methods were proposed. The model doesn't show enough novelty in methodology, and the experimental validation has flaws as detailed below.

Response: We deeply appreciate your insightful comments. Different from the previous methods, the DeepGlycanSite has a geometry-enhanced equivariant graph neural network with vector-scalar interactive updating units and considers the evolutionary features, contributing to its outstanding performance. We compared the DeepGlycanSite with the general binding site prediction method (LigBind), two re-trained modern models and two re-implemented traditional machine learning methods with the glycan binding dataset. DeepGlycanSite still outperformed these alternative methods. We carried out ablations to interpret the network architecture of DeepGlycanSite, and performed cross validations (CV) to demonstrate its robustness. The influence of datasets redundancy on the model was also investigated and discussed. For experimental validation, we conducted more mutagenesis studies for both positive and negative predictions of our model. We are confident that these substantial revisions and the point-by-point responses provided herein address your concerns.

1. The methodology doesn't show enough novelty. Many recent techniques like equivalent GCN has shown improvements over this employed original GCN architecture. Though the method was shown to outperform other methods like Pesto, these methods themselves were designed for binding site with ligands, not for glycans. It's interesting to re-train these modern models on this glycan binding dataset or re-implement traditional ML methods to see the performance. Alternatively, they can indicate the method on the well-studied PPI or DNA binding sites prediction. On the other hand, the authors can compare with the recently developed general binding site (PMID: 37054909).

Response: We appreciate your valuable comments. Accordingly, we compared our methods with the recently developed general binding site prediction method, i.e. LigBind (PMID: 37054909), two re-trained modern models and two re-implemented traditional machine learning methods. In addition, we carried out ablation experiments to interpret the network architectures of our models.

First, we used the LigBind to predict the glycan-binding sites on the independent dataset T145. Since these ligands are not included in the LigBind library (1159 ligands), we employed LigBind-G and LigBind-G-NO for prediction (<https://github.com/YYingXia/LigBind/>). As shown in the **Table R1**, DeepGlycanSite outperformed the alternative methods with respect to the Matthews correlation coefficient (MCC), precision and balanced accuracy.

Second, we re-trained two modern models (PeSTo and DeepPocket) on our glycan binding dataset, following their protocols (PMIDs: 37072397 and 34374539). On the independent

dataset T145, DeepGlycanSite showed remarkably better performance on the glycan-binding site prediction with respect to MCC and precision, compared with these re-trained modern models (**Table R2**). DeepPocket uses a voxel-based convolutional neural network (CNN) to predict ligand-binding pockets. Different from DeepPocket, DeepGlycanSite employs an equivariant graph neural network (EGNN), handling the sparse connections among nodes of the network. Both DeepGlycanSite and PeSTo are EGNN models converting geometric information into vectors and scalars for feature representation. Nevertheless, DeepGlycanSite leverages a more sophisticated message passing architecture, which makes full use of vectors and scalars by updating both node and edge features (**Fig. R1**). But PeSTo only updates node features after message passing. In addition to the geometric information, DeepGlycanSite considers the evolutionary features of the protein. Both enhanced EGNN and the evolutionary features may contribute to the performance gains of DeepGlycanSite. To validate this assumption, we carried out ablation experiments of the DeepGlycanSite. As shown in **Table R3**, the elimination of the vector-scalar interactions in the updating units remarkably attenuated the performance of the DeepGlycanSite. Meanwhile, deleting the evolutionary features also led to inferior performance. In brief, the vector-scalar interactions and evolutionary features are indispensable in DeepGlycanSite. We also conducted ablation experiments of the DeepGlycanSite_{+Ligand} on the independent testing set TM175 (**Table R4**), indicating the ligand information, especially the ligand vector, as a key component for DeepGlycanSite_{+Ligand}.

To compare our model with traditional machine learning (ML) methods, we have further re-implemented two models, i.e. support vector machines (SVM) and extreme gradient boosting (XGBoost), on our glycan binding dataset. We extracted feature vectors with 1309

dimensions from our training dataset. The labeling of each amino acid residue was consistent with its classification in DeepGlycanSite. To preserve the integrity of protein data, residues from the same protein were allocated to the same batch for training purposes. For the SVM model, we employed the scikit-learn library, configured with a hinge loss function. In the case of the XGBoost model, we used the XGBoost Python package, setting the hyperparameters to include a maximum depth of 6, a learning rate of 0.3, with the objective set to 'binary:logistic' and the evaluation metric to 'logloss'. DeepGlycanSite significantly outperformed the SVM model and the XGBoost model (**Table R5**). Compared with these two ML models, DeepGlycanSite used the graph representations to effectively capture the correlations between protein residues, contributing to the performance gains.

In conclusion, DeepGlycanSite outperformed the recently developed general binding site prediction method (LigBind), two re-trained modern models (PeSTo and DeepPocket) and two re-implemented traditional ML models (SVM and XGBoost). Different from these alternative methods, the DeepGlycanSite has an enhanced EGNN with sophisticated vector-scalar interactive updating units. Incorporation of evolutionary features also helps improve its performance. These findings underscore the novelty of our method. We have included this information in the revised manuscript.

Table R1 | Comparing DeepGlycanSite with LigBind models on the independent dataset T145.

Method	MCC	Precision	Balanced Accuracy
LigBind-G	0.051 ± 0.662 ^{***}	0.464 ± 0.417 ^{**}	0.624 ± 0.127 ^{***}
LigBind-G-NO	0.183 ± 0.496 ^{***}	0.440 ± 0.361 ^{***}	0.628 ± 0.115 ^{***}
DeepGlycanSite	0.625 ± 0.292	0.631 ± 0.306	0.829 ± 0.156

Data represent means ± standard deviation. Two-tailed Mann-Whitney U test is used to determine statistical difference between DeepGlycanSite and an alternative method. ** indicates P is less than 0.01, *** indicates P is less than 0.001.

Table R2 | Comparing DeepGlycanSite with two re-trained modern models on the independent dataset T145.

Method	MCC	Precision	Balanced accuracy
PeSTo (re-trained)	0.423 ± 0.272 ^{***}	0.331 ± 0.238 ^{***}	0.813 ± 0.168
DeepPocket (re-trained)	0.297 ± 0.520 ^{***}	0.310 ± 0.249 ^{***}	0.778 ± 0.211
DeepGlycanSite	0.625 ± 0.292	0.631 ± 0.306	0.829 ± 0.156

Data represent means ± standard deviation. Two-tailed Mann-Whitney U test is used to determine statistical difference between DeepGlycanSite and an alternative method. *** indicates P is less than 0.001.

Table R3 | Ablation results of DeepGlycanSite on the independent dataset T145.

ID	Ablated item	MCC	Precision	Balanced accuracy
1	Geometric features	0.608 ± 0.269	$0.569 \pm 0.276^*$	0.845 ± 0.148
2	Evolutionary features	$0.559 \pm 0.279^{**}$	$0.532 \pm 0.287^{**}$	0.816 ± 0.157
3	Scalar-vector interactions in the updating units	$0.420 \pm 0.246^{***}$	$0.399 \pm 0.234^{***}$	$0.746 \pm 0.145^{***}$
4	Transformer	0.570 ± 0.331	$0.567 \pm 0.294^*$	0.840 ± 0.158
5	None	0.625 ± 0.292	0.631 ± 0.306	0.829 ± 0.156

Data represent means \pm standard deviation. Two-tailed Mann-Whitney U test is used to determine statistical difference between DeepGlycanSite and ablated DeepGlycanSite. * indicates P is less than 0.05, ** indicates P is less than 0.01, *** indicates P is less than 0.001.

Table R4 | Ablation results of DeepGlycanSite+Ligand on the independent dataset TM175.

ID	Ablated item	MCC	Precision	Balanced accuracy
1	Molecule features	0.500 ± 0.322	0.482 ± 0.318	0.777 ± 0.177
2	Ligand vector	$0.415 \pm 0.310^{***}$	$0.390 \pm 0.291^{***}$	$0.744 \pm 0.184^{**}$
3	Ligand graph	0.495 ± 0.313	0.503 ± 0.307	$0.797 \pm 0.174^*$
4	All ligand information	$0.399 \pm 0.324^{***}$	$0.380 \pm 0.318^{***}$	$0.731 \pm 0.182^{***}$
5	None	0.538 ± 0.321	0.504 ± 0.327	0.806 ± 0.163

Data represent means \pm standard deviation. Two-tailed Mann-Whitney U test is used to determine statistical difference between DeepGlycanSite+Ligand and ablated

DeepGlycanSite+Ligand. * indicates P is less than 0.05, ** indicates P is less than 0.01, *** indicates P is less than 0.001.

Table R5 | Comparing DeepGlycanSite with two re-implemented machine learning models on the independent dataset T145.

Method	MCC	Precision	Balanced accuracy
SVM	$0.068 \pm 0.544^{***}$	$0.408 \pm 0.353^{***}$	$0.597 \pm 0.098^{***}$
XGBoost	$0.126 \pm 0.470^{***}$	$0.429 \pm 0.361^{***}$	$0.588 \pm 0.095^{***}$
DeepGlycanSite	0.625 ± 0.292	0.631 ± 0.306	0.829 ± 0.156

Data represent means \pm standard deviation. Two-tailed Mann-Whitney U test is used to determine statistical difference between DeepGlycanSite and an alternative method. *** indicates P is less than 0.001.

Fig. R1 | The comprehensive attention layer of ReceptorNet. n is the node feature, e is the edge feature, \vec{v}_i (or \vec{v}_j) and \vec{v}_e are node and edge vectors, respectively. Red circle labels the edge updating according to vectors. Different operations are represented by different arrows: blue arrows are projection, gray arrows are arithmetic computation, green arrows are splitting, and purple arrows are rejection.

2. For the current dataset, the authors only control sequence identity of 30% (generally should be 25%) between training/test set but not excluding redundant sequences within training/test sets. This may cause bias toward some popular protein family. In addition, a CV on the training set is also needed to indicate the robustness of the method.

Response: We appreciate your valuable comments. Accordingly, we re-trained and then evaluated our models using training/testing sets with different protein sequence identities. And we also performed a CV to validate the robustness of our method.

In general, carbohydrate binding specificity could result from limited amino acid changes within a common protein fold. For example, the calcium-dependent lectins possess common carbohydrate-recognition domains with subtly different carbohydrate-binding motifs. Such lectins with the Glu-Pro-Asn (EPN) motifs bind to mannoses, while those with Gln-Pro-Asp (QPD) motifs bind to galactoses (PMIDs: 28374848 and 29718486). In addition, the same protein domain could bind to different carbohydrates (PMIDs: 37955640, 37527445, 38602480). Therefore, it is important to include homologous (or even same) proteins binding to different carbohydrates. To reduce possible bias toward some popular proteins and consider more protein-carbohydrate interactions, for the current training set, we only excluded any instance of the same site binding to the same carbohydrate. For the current testing set, any protein with more than 95% sequence identity to those of training/validation sets was excluded. We further controlled protein sequence identity of 30% within the testing set to construct a nonredundant independent dataset T145. The T145 involves 145 carbohydrate-protein complexes.

To investigate the effect of the protein redundancy on the training set, we excluded the homologous proteins (with the sequence identity more than 95%) within the current training set to construct a new training set, termed Train_{NR95}. We used the Train_{NR95} to re-train our model, and then compared with the current DeepGlycanSite model on the independent testing set T145. As shown in **Table R6**, the re-trained model did not show better performance compared with the current DeepGlycanSite model. In addition, we controlled sequence identity of 25% between the testing set (T145) and the training/validation sets to construct a new independent testing set T59. T59 involved 59 carbohydrate-protein complexes. As shown in **Table R7**, DeepGlycanSite still outperformed the alternative methods with respect to MCC and precision on the T59 dataset.

Moreover, we conducted CV experiments for DeepGlycanSite and DeepGlycanSite+Ligand, respectively. Both models showed consistent performance (**Tables R8 and R9**), affirming their reliability and effectiveness in predicting carbohydrate-binding sites. We have revised the manuscript to include these results.

Table R6 | Comparing the baseline and re-trained DeepGlycanSite models on the independent dataset T145.

Model	MCC	Precision	Balanced accuracy
DeepGlycanSite (re-trained)	0.588 ± 0.281	0.590 ± 0.300	0.814 ± 0.153
DeepGlycanSite (baseline)	0.625 ± 0.292	0.631 ± 0.306	0.829 ± 0.156

The re-trained DeepGlycanSite model used a new training set, i.e., Train_{NR95}, including proteins with less than 95% sequence identity. Data represent means ± standard deviation.

Table R7 | Comparing DeepGlycanSite with alternative methods on the independent dataset T59.

Method	MCC	Precision	Balanced accuracy
StackCBPred	0.017 ± 0.085 ^{***}	0.050 ± 0.034 ^{***}	0.523 ± 0.093 ^{***}
Fpocket	0.190 ± 0.324 ^{***}	0.195 ± 0.281 ^{***}	0.617 ± 0.198 ^{***}
SiteMap	0.169 ± 0.452 ^{***}	0.187 ± 0.199 ^{***}	0.698 ± 0.214
DeepPocket	0.287 ± 0.476 ^{***}	0.288 ± 0.227 ^{***}	0.760 ± 0.207
PeSTo	0.258 ± 0.349 ^{***}	0.204 ± 0.148 ^{***}	0.769 ± 0.172
DeepGlycanSite	0.532 ± 0.299	0.549 ± 0.302	0.778 ± 0.161

Data represent means ± standard deviation. Two-tailed Mann-Whitney U test is used to determine statistical difference between DeepGlycanSite and an alternative method. ^{***} indicates P is less than 0.001.

Table R8 | Five-fold cross validation of DeepGlycanSite on the dataset T145.

Fold	MCC	Precision	Balanced Accuracy
1	0.609 ± 0.276	0.592 ± 0.281	0.833 ± 0.153
2	0.603 ± 0.288	0.600 ± 0.300	0.823 ± 0.156
3	0.591 ± 0.284	0.621 ± 0.303	0.802 ± 0.156
4	0.625 ± 0.292	0.631 ± 0.306	0.829 ± 0.156
5	0.596 ± 0.304	0.623 ± 0.322	0.807 ± 0.165
Average	0.605 ± 0.012	0.613 ± 0.015	0.819 ± 0.012

Data represent means ± standard deviation.

Table R9 | Five-fold cross validation of DeepGlycanSite_{+Ligand} on the dataset TM175.

Fold	MCC	Precision	Balanced Accuracy
1	0.527 ± 0.325	0.498 ± 0.326	0.795 ± 0.167
2	0.538 ± 0.321	0.504 ± 0.327	0.806 ± 0.163
3	0.536 ± 0.326	0.552 ± 0.344	0.775 ± 0.161
4	0.534 ± 0.323	0.493 ± 0.321	0.804 ± 0.169
5	0.530 ± 0.325	0.518 ± 0.334	0.787 ± 0.165
Average	0.533 ± 0.004	0.513 ± 0.023	0.793 ± 0.012

Data represent means ± standard deviation.

3. Current wet experimental didn't validate their model. First only one out of 8 predicted showed significantly decreased binding affinity, indicating the poor prediction binding site. Secondly, they only mutated predicted binding residues but didn't employ other residues as control. The drop in the binding affinity can be caused by direct interaction, and also by the impact on the overall structure. They can select to mutate AA in surrounding residues (like #286) and residues in other regions as a contrast, or they can determine the complex structure.

Response: We appreciate your valuable comments. Accordingly, we mutated predicted binding residues and employed other residues as control (like #286) for experimental validations. Using DeepGlycanSite+Ligand and AlphaFold2-predicted human P2Y14 structure, we identified eight possible fucose-moiety-binding residues (G80, D81, N90, V93, Q169, I170, A285 and N287), whose GDP-binding probability is less than 0.2 and GDP-Fuc-binding probability is more than 0.5 (**Fig. R2a-b**). To validate the prediction, we designed single-point mutations for these eight predicted binding residues and four surrounding residues (L79, V91, F92 and A286). Among them, single-point mutations of four predicted binding residues (G80, D81, N90 and V93) significantly reduced the GDP-Fuc-induced receptor activities (**Fig. R2c** and **Table R10**). The potencies of GDP-Fuc reduced by five- to ten-fold on three mutants (G80R, D81A and V93M), compared with that in the WT group (**Fig. R2c** and **Table R10**). In contrasts, these three mutations did not show great impacts on reduction of GDP-induced receptor activities (**Fig. R2c** and **Table R11**). These results validate that G80, D81 and V93 are key residues for fucose moiety-recognition of P2Y14. Meanwhile, substitute of N287 by alanine diminished GDP-Fuc and GDP-induced responses (**Fig. R3**, **Tables R11** and **R12**).

Other residues, especially four surrounding predicted nonbinding ones, did not exhibit great impact on receptor activation by GDP-Fuc or GDP (**Fig. R3, Tables R10 and R11**). These results not only validate our assumption that GDP-Fuc directly binds to and acts on P2Y14 but also identify the fucose moiety-recognition residues. We have included these results in the revised manuscript.

Fig. R2 | Experiment validation of DeepGlycanSite_{+Ligand}. **a** Predicted binding sites of GDP (green) and GDP-Fuc (orange) by DeepGlycanSite_{+Ligand} on the AlphaFold2-predicted human P2Y14 structure. **b** Binding probabilities of predicted carbohydrate-binding sites for GDP and GDP-Fuc, respectively. The predicted fucose-binding residues are labelled with red arrows. **c** Calcium mobilization concentration-response curves for GDP or GDP-Fuc in HEK293 expressing P2Y14 WT and mutants ($n = 3$).

Fig. R3 | Calcium mobilization concentration-response curves for GDP or GDP-Fuc in HEK293 expressing P2Y14 WT and mutants ($n = 3$).

Table R10 | The EC₅₀ values for P2Y₁₄ and its mutant activation upon GDP-Fuc binding in the calcium mobilization assays.

Construct	EC₅₀ (μM)	n	Statistics	Comment
WT	0.49 ± 0.04	12	T.TEST	
L79A	0.54 ± 0.14	3	NS	WT vs. L79A
G80A	0.94 ± 0.11	4	P < 0.001	WT vs. G80A
G80R	3.31 ± 0.27	6	P < 0.0001	WT vs. G80R
D81A	5.22 ± 0.49	6	P < 0.0001	WT vs. D81A
N90A	1.02 ± 0.03	3	P < 0.0001	WT vs. N90A
V91A	0.61 ± 0.15	3	NS	WT vs. V91A
V91R	0.62 ± 0.19	3	NS	WT vs. V91R
F92M	0.58 ± 0.09	3	NS	WT vs. F92M
V93A	1.39 ± 0.13	6	P < 0.0001	WT vs. V93A
V93M	4.13 ± 0.58	5	P < 0.0001	WT vs. V93M
Q169A	0.51 ± 0.04	5	NS	WT vs. Q169A
I170A	0.85 ± 0.07	5	P < 0.001	WT vs. I170A
A285G	0.45 ± 0.04	5	NS	WT vs. A285G
A286G	0.48 ± 0.03	3	NS	WT vs. A286G
A286L	0.45 ± 0.05	3	NS	WT vs. A286L
N287A	Not detectable	3	-	WT vs. N287A

NS indicates the difference is not statistically significant.

Table R11 | The EC₅₀ values for P2Y₁₄ and its mutant activation upon GDP binding in the calcium mobilization assays.

Construct	EC₅₀ (μM)	n	Statistics	Comment
WT	0.24 ± 0.01	10	T.TEST	
L79A	0.26 ± 0.06	3	NS	WT vs. L79A
G80A	0.35 ± 0.02	3	P < 0.001	WT vs. G80A
G80R	0.10 ± 0.01	6	P < 0.0001	WT vs. G80R
D81A	0.17 ± 0.01	6	P < 0.001	WT vs. D81A
N90A	0.22 ± 0.03	3	NS	WT vs. N90A
V91A	0.24 ± 0.02	3	NS	WT vs. V91A
V91R	0.23 ± 0.03	3	NS	WT vs. V91R
F92M	0.27 ± 0.04	3	NS	WT vs. F92M
V93A	0.41 ± 0.06	3	P < 0.001	WT vs. V93A
V93M	0.40 ± 0.02	6	P < 0.0001	WT vs. V93M
Q169A	0.27 ± 0.00	3	NS	WT vs. Q169A
I170A	0.31 ± 0.02	3	P < 0.05	WT vs. I170A
A285G	0.20 ± 0.03	3	NS	WT vs. A285G
A286G	0.26 ± 0.01	3	NS	WT vs. A286G
A286L	0.26 ± 0.04	3	NS	WT vs. A286L
N287A	Not detectable	3	-	WT vs. N287A

NS indicates the difference is not statistically significant.

Reviewer #2

In this paper, the researchers introduce DeepGlycanSite, a deep learning model designed to forecast carbohydrate binding sites across diverse proteins. Additionally, they conducted a comparative analysis with existing software for detecting carbohydrate binding sites, demonstrating superior performance using their proposed method. They also applied the method to human P2Y14 and confirmed their results with experimental observations.

1. In the introduction, include that a challenge associated with the utilization of carbohydrate-based drugs also lies in their specificity. This limitation arises not only due to a lack of comprehension regarding their binding mechanisms but also because even when such understanding exists, achieving the desired specificity proves to be a challenging task.

Response: We agree with the reviewer. It is challenging to achieve the desired specificity of a carbohydrate on the therapeutic target (*Nat Rev Drug Discov* 2009, 8:661). We have included this information in the introduction.

2. If I grasp the information correctly, you employed PDB structures in training the model, where the input consists of protein structures from which carbohydrates were extracted. The test set also employed a similar structure type. My concern stems from the potential bias in such a structure, wherein the mere presence of a carbohydrate within it might influence the

model to identify the attachment site. This bias arises from the protein structure remaining unaltered compared to the one with the attached carbohydrate. Can you comment on this?

Response: We appreciate your insightful comments. We constructed the training, validation and testing sets based on a large carbohydrate-protein complex dataset. Therefore, only carbohydrate-bound protein structures are employed in training and testing of DeepGlycanSite.

Since the protein is a flexible molecule, its carbohydrate-bound (holo) structure could be different from the carbohydrate-free (apo) one. The differences between holo and apo structures have been widely studied. In general, the main-chain structural difference is small between holo and apo structures, although the side-chain arrangement could largely differ (*J Chem Inf Model* 2021, 61:535). Gutteridge and Thornton analyzed 60 enzyme molecules and found that the C α RMSDs between apo and holo structures for most enzymes were less than 1 Å (*J Mol Biol* 2005, 346:21). Clark et al. investigated more than 4000 crystal structures of 350 proteins and found that the main-chain motion of apo-holo structures is small, but larger differences occur in side-chain orientations (*PLoS Comput Biol* 2019, 15: e1006705).

DeepGlycanSite exploits geometric features, including intra- and inter-residue orientations and distances, as well as evolutionary information to present proteins in graph representations at the residual level. It considers geometric information involving main-chain, C β and C γ atoms. Therefore, for the same protein, the prediction of the apo structure may differ from that of the holo structure when using DeepGlycanSite. We have developed new testing sets composed of apo structures, using AlphaFold predictions and molecular dynamics (MD) simulations, to further evaluate the capabilities of DeepGlycanSite.

3. *Have you attempted to assess your model using various structures of the same protein?*

Testing the model across diverse structures would serve as a robust validation, especially considering the exclusion of proteins with more than 30% homology from both the training and testing sets. Achieving consistent results across multiple structures would strongly affirm the effectiveness of your model.

Response: We excluded proteins with more than 25% homology from both the training and testing sets to construct a new independent testing set T59. T59 involved 59 unique carbohydrate-binding proteins. As shown in **Table R12**, DeepGlycanSite outperformed the alternative methods on the T59 dataset. To obtain various structures of the same protein, we employed AlphaFold2 and AlphaFold2 Multimer (*Nature* 2021, 596:583; *bioRxiv* 2022, 2021.2010.2004.463034) to predict protein structures based on the protein sequences of T59. The top five ranked conformation models for each protein were selected to construct a new independent testing set T59_{AF2}, consisting of 59 unique proteins and 295 apo structure models. As shown in **Table R13**, DeepGlycanSite remarkably outperformed the alternative methods on the T59_{AF2}, demonstrating its robustness and effectiveness. We have included this information in the revised manuscript.

Table R12 | Comparing DeepGlycanSite with alternative methods on the dataset T59.

Method	MCC	Precision	Balanced accuracy
StackCBPred	$0.017 \pm 0.085^{***}$	$0.050 \pm 0.034^{***}$	$0.523 \pm 0.093^{***}$
Fpocket	$0.190 \pm 0.324^{***}$	$0.195 \pm 0.281^{***}$	$0.617 \pm 0.198^{***}$
SiteMap	$0.169 \pm 0.452^{***}$	$0.187 \pm 0.199^{***}$	0.698 ± 0.214
DeepPocket	$0.287 \pm 0.476^{***}$	$0.288 \pm 0.227^{***}$	0.760 ± 0.207
PeSTo	$0.258 \pm 0.349^{***}$	$0.204 \pm 0.148^{***}$	0.769 ± 0.172
DeepGlycanSite	0.532 ± 0.299	0.549 ± 0.302	0.778 ± 0.161

Data represent means \pm standard deviation. Two-tailed Mann-Whitney U test is used to determine statistical difference between DeepGlycanSite and an alternative method. *** indicates P is less than 0.001.

Table R13 | Comparing DeepGlycanSite with alternative methods on the dataset T59_{AF2}.

Method	MCC	Precision	Balanced Accuracy
Fpocket	$-0.048 \pm 0.476^{***}$	$0.135 \pm 0.189^{***}$	$0.571 \pm 0.123^{***}$
SiteMap	$0.233 \pm 0.241^{***}$	$0.172 \pm 0.170^{***}$	$0.690 \pm 0.189^*$
DeepPocket	$-0.488 \pm 0.669^{***}$	$0.131 \pm 0.230^{***}$	$0.590 \pm 0.154^{***}$
PeSTo	$0.289 \pm 0.270^{***}$	$0.220 \pm 0.165^{***}$	0.762 ± 0.161
DeepGlycanSite	0.467 ± 0.272	0.436 ± 0.274	0.777 ± 0.153

Data represent means \pm standard deviation. Two-tailed Mann-Whitney U test is used to determine statistical difference between DeepGlycanSite and an alternative method. * indicates P is less than 0.05, *** indicates P is less than 0.001.

4. Conversely, the AlphaFold model underwent equilibration, and a representative structure was selected following a simulation lasting 300-500 ns. While the binding site prediction was accurate, it's worth noting that this site corresponds to a region of the GPCR protein that other software similarly identifies as a binding site, primarily due to its proximity to the orthosteric binding site.

Response: We agree with you. The fucose moiety-recognition residues are proximity to the orthosteric binding site.

5. Have you tried with conducting MD simulations on a select set of representative examples and subsequently applied your protocol to assess a few representative structures? This approach would help determine the robustness of your method against conformational changes. Additionally, such testing could mitigate the bias introduced by the presence of carbohydrates in the structure. I would strongly suggest this.

Response: Thanks for your valuable comments. Accordingly, we conducted 20 ns MD simulations for all proteins in the T59 dataset. Prior to conducting MD simulations, we removed all hydrogens and reintroduced them via the tleap module in Amber. The protonation state of each residue was determined using PROPKA3 (*J Chem Theory Comput* 2011, 7: 525), while disulfide bonds were manually configured. The FF19SB force field was selected for protein modeling (*J Chem Theory Comput* 2020, 16: 528). Each complex was initially solvated

in an orthorhombic transferable intermolecular potential three-point (TIP3P) water box (*J Chem Phys* 1983, 79: 926). To neutralize the system and simulate physiological conditions, counterions and an additional 0.15 mol/L NaCl solution were added. Subsequently, the systems underwent a two-stage energy minimization process. In the first stage, the proteins were held in place with a positional restraint of 500 kcal/mol·Å², allowing other molecules, including water and counterions, to be minimized over 2000 steps of steepest descent minimization, followed by 3000 steps of conjugate gradient minimization. In the second stage, the entire systems were minimized without constraints, utilizing the steepest descent method for 4,000 cycles and the conjugate gradient method for an additional 6,000 cycles. Post-minimization, systems were gradually heated from 0 to 300 K over 300 ps, with the complexes under a positional restraint of 10 kcal/mol·Å² within a canonical ensemble (NVT). This was followed by a 700 ps NVT equilibration period at 300 K, maintaining a positional restraint of 10 kcal/mol·Å² on the proteins. Thereafter, 20 ns MD simulations were conducted at 1 atm pressure and 300 K temperature using pmemd.cuda in AMBER 20 software suite. System temperatures were regulated via Langevin dynamics, with a collision frequency of 1 ps⁻¹. Long-range electrostatic interactions were addressed using the Particle Mesh Ewald (PME) method (*J Chem Phys* 1993, 98: 10089), with a cutoff of 10 Å for managing short-range electrostatic and van der Waals forces. The PME grid spacing was maintained at 1 Å. SHAKE algorithm was utilized to constrain bonds containing hydrogen (*J Comput Phys* 1977, 23: 327). For representative structure selection, we clustered the MD simulation trajectories into five distinct groups using the Amber CPPTRAJ “cluster” program with the hierarchical agglomerative algorithm (*J Chem Theory Comput* 2013, 9: 3084; *J Chem Theory Comput* 2007, 3: 2312). The

distances between two different structures were calculated as the best-fit coordinate RMSD using all heavy atoms. The structure whose RMSD is smallest to the others were selected as the representative one. Each protein has five representative structures. We collected all representative structures to construct a new independent testing set T59_{MD}, consisting of 59 unique proteins and 295 apo structure models. DeepGlycanSite remarkably outperformed the alternative methods on the T59_{MD} (**Table R14**), underscoring its robustness and reliability against conformational changes.

Table R14 | Comparing DeepGlycanSite with alternative methods on the dataset T59_{MD}.

Method	MCC	Precision	Balanced Accuracy
Fpocket	-0.148 ± 0.389 ^{***}	0.055 ± 0.067 ^{***}	0.513 ± 0.034 ^{***}
SiteMap	0.106 ± 0.122 ^{***}	0.097 ± 0.090 ^{***}	0.592 ± 0.099 ^{***}
DeepPocket	0.035 ± 0.215 ^{***}	0.100 ± 0.082 ^{***}	0.577 ± 0.082 ^{***}
PeSTo	0.040 ± 0.191 ^{***}	0.095 ± 0.082 ^{***}	0.573 ± 0.091 ^{***}
DeepGlycanSite	0.431 ± 0.255	0.412 ± 0.250	0.751 ± 0.144

Data represent means ± standard deviation. Two-tailed Mann-Whitney U test is used to determine statistical difference between DeepGlycanSite and an alternative method. *** indicates *P* is less than 0.001.

Reviewer #3

The authors developed a deep learning model DeepGlycanSite to predict carbohydrate-binding sites. The PDB dataset was used for training and validation after removal of homologous proteins. The performance of DeepGlycanSite is better than the previous methods including DeepPocket and PeSTo which are also based on deep learning methods. Finally, the authors experimentally validated the prediction of DeepGlycanSite on a GPCR P2Y14.

1. In general, subtle differences in carbohydrate binding specificity result from limited amino acid changes in the ligand binding site of a same protein fold. For example, EPN and QPD motifs are presented on a common C-type lectin fold but bind different sugars, Man and Gal. Legume lectins share a common fold but have different binding specificities. The authors omitted homologous proteins for training and validation and I am afraid that such subtle differences in glycan ligands may not be considered in DeepGlycanSite. In other words, proteins of a same protein fold can bind to different sugars. Typical lectin folds are C-type lectins, legume lectins, and I-type lectins etc. This point needs to be validated and discussed.

Response: We highly agree with you. Proteins of a same protein fold can bind to different sugars. Therefore, we indeed included homologous proteins for training and validation, and considered subtle differences in glycan ligands in DeepGlycanSite. To include more protein-carbohydrate interactions and reduce possible bias toward some popular proteins, for the

training and validation, we only excluded any instance of the same site binding to the same carbohydrate. We have revised the manuscript to include this information for clarity.

2. Relatively good performance was seen for predicting nucleotide binding. Did DeepGlycanSite distinguish the sugar nucleotides, such as GDP-Fuc and GDP-Man that share a common nucleotide but have different sugars? For example, how was the prediction of GDP-Man binding to P2Y14?

Response: We employed DeepGlycanSite_{+Ligand} to predict the binding site of GDP-Man on the P2Y14, using the AlphaFold2-predicted structure model. As shown in **Fig. R4**, DeepGlycanSite_{+Ligand} predicted that the GDP-Man and GDP-Fuc shared common binding residues for their sugar moieties. We have validated four predicted-binding residues (G80, D81, N90 and V93) as the fucose-moiety recognition site in the calcium mobilization assays (**Fig. 5**). Among these four residues, D81 and N90 had GDP-Man-binding probabilities more than 0.5, but the other two had probabilities less than 0.5 (0.403 for G80 and 0.398 for V93). In brief, there are only subtle differences between the prediction of GDP-Man and prediction of GDP-Fuc binding to P2Y14.

Fig. R4 | Binding probabilities of predicted carbohydrate-binding sites for GDP, GDP-Fuc and GDP-Man, respectively. The predicted fucose-binding residues are labelled with red arrows.

3. *DeepPocket and PeSTo use deep learning methods but DeepGlycanSite showed better performance than these methods. What makes DeepGlycanSite different from these methods?*

Response: DeepPocket uses a voxel-based convolutional neural network to predict ligand-binding pockets. Different from DeepPocket, DeepGlycanSite employs an equivariant graph neural network (EGNN), handling the sparse connections among nodes of the network. Both DeepGlycanSite and PeSTo are EGNN models converting geometric information into vectors and scalars for feature representation. Nevertheless, DeepGlycanSite leverages a more sophisticated message passing architecture, which makes full use of vectors and scalars by updating both node and edge features (**Fig. R5**). But PeSTo only updates node features after message passing. In addition to the geometric information, DeepGlycanSite considers the evolutionary features of the protein. Both enhanced EGNN and the evolutionary features may contribute to the performance gains of DeepGlycanSite. To validate this assumption, we carried out ablation experiments of the DeepGlycanSite. As shown in **Table R15**, the elimination of the vector-scalar interactions in the updating units remarkably attenuated the performance of the DeepGlycanSite. Meanwhile, deleting the evolutionary features also led to inferior performance. In brief, the vector-scalar interactions and evolutionary features are indispensable in DeepGlycanSite. We also conducted ablation experiments of the DeepGlycanSite+Ligand on the independent testing set TM175 (**Table R16**), indicating the ligand information, especially the ligand vector, as a key component for the model.

Fig. R5 | The comprehensive attention layer of ReceptorNet. n is the node feature, e is the edge feature, \vec{v}_i (or \vec{v}_j) and \vec{v}_e are node and edge vectors, respectively. Red circle labels the edge updating according to vectors. Different operations are represented by different arrows: blue arrows are projection, gray arrows are arithmetic computation, green arrows are splitting, and purple arrows are rejection.

Table R15 | Ablation results of DeepGlycanSite on the independent dataset T145.

ID	Ablated item	MCC	Precision	Balanced accuracy
1	Geometric features	0.608 ± 0.269	$0.569 \pm 0.276^*$	0.845 ± 0.148
2	Evolutionary features	$0.559 \pm 0.279^{**}$	$0.532 \pm 0.287^{**}$	0.816 ± 0.157
3	Scalar-vector interactions in the updating units	$0.420 \pm 0.246^{***}$	$0.399 \pm 0.234^{***}$	$0.746 \pm 0.145^{***}$
4	Transformer	0.570 ± 0.331	$0.567 \pm 0.294^*$	0.840 ± 0.158
5	None	0.625 ± 0.292	0.631 ± 0.306	0.829 ± 0.156

Data represent means \pm standard deviation. Two-tailed Mann-Whitney U test is used to determine statistical difference between DeepGlycanSite and ablated DeepGlycanSite. * indicates P is less than 0.05, ** indicates P is less than 0.01, *** indicates P is less than 0.001.

Table R16 | Ablation results of DeepGlycanSite_{+Ligand} on the independent dataset TM175.

ID	Ablated item	MCC	Precision	Balanced accuracy
1	Molecule features	0.500 ± 0.322	0.482 ± 0.318	0.777 ± 0.177
2	Ligand vector	$0.415 \pm 0.310^{***}$	$0.390 \pm 0.291^{***}$	$0.744 \pm 0.184^{**}$
3	Ligand graph	0.495 ± 0.313	0.503 ± 0.307	$0.797 \pm 0.174^*$
4	All ligand information	$0.399 \pm 0.324^{***}$	$0.380 \pm 0.318^{***}$	$0.731 \pm 0.182^{***}$
5	None	0.538 ± 0.321	0.504 ± 0.327	0.806 ± 0.163

Data represent means \pm standard deviation. Two-tailed Mann-Whitney U test is used to determine statistical difference between DeepGlycanSite_{+Ligand} and ablated

DeepGlycanSite+Ligand. * indicates P is less than 0.05, ** indicates P is less than 0.01, *** indicates P is less than 0.001.

4. The description of the MD simulation of the P2Y14 complex is very poor and I could not find out how the representative model was selected. The relevant data and criteria must be presented. Otherwise, the model cannot be treated as a reliable model even though experimental results are available.

Response: We thank your valuable comments. Accordingly, we provided more description of the MD simulation of the P2Y14 complex in the revised manuscript.

Initially, a ligand was placed in the center of the predicted binding site and minimized using Schrödinger's Maestro to avoid conflict. To build a simulation system, we placed the complex model into a 1-palmitoyl-2-oleoyl-sn-glycero-3-phosphocholine lipid bilayer. The lipid embedded complex model was solvated in a periodic boundary condition box ($60 \text{ \AA} \times 60 \text{ \AA} \times 135 \text{ \AA}$) filled with TIP3P water molecules (*J Comput Phys* 1983, 79: 926) and 0.15 M NaCl using CHARMM-GUI (*J Comput Chem* 2014, 35: 1997). Each system was replicated to perform three independent simulations for 500 ns. On the basis of the CHARMM36m all-atom force field (*Nat Methods* 2017, 14: 71) for protein and CHARMM general force field (*J Comput Chem* 2010, 31: 671) for ligand, molecular dynamics simulations were conducted using Amber20 pmemd.cuda (*J Chem Theory Comput* 2013, 9: 3878). After minimization process of 5000 steepest descent cycles with a constraint on backbone atom, sidechain atom, and lipid coordinates and a constraint on dihedrals, the constraints were generally decreased in the

separated 6 steps of the equilibration process provided by CHARMM-GUI (*J Comput Chem* 2014, 35: 1997). 500 ns production run was then carried out for each simulation. All productions were carried out in the NPT ensemble at temperature of 303.15 K and a pressure of 1 atm. Temperature and pressure were controlled using the Nose-Hoover thermostat (*J Chem Phys* 1985, 83: 4069) and the Parrinello–Rahman barostat (*J Appl Phys* 1981, 52: 7182), respectively. Equations of motion were integrated with a 2-fs time step as the SHAKE algorithm was used to constrain bond length (*J Comput Phys* 1977, 23: 327). Long-range electrostatic interactions were addressed using the Particle Mesh Ewald method (*J Chem Phys* 1993, 98: 10089). Short-range electrostatic and van der Waals interactions were treated with a 12 Å cutoff, which were gradually switched off between 12 Å and 10 Å. The final 200 ns trajectory of each simulation was used for the extraction of representative structure.

For representative model selection, we clustered the MD simulation trajectories using the Amber CPPTRAJ “cluster” program with the hierarchical agglomerative algorithm (*J Chem Theory Comput* 2013, 9: 3084; *J Chem Theory Comput* 2007, 3: 2312). The distances between two different models were calculated as the best-fit coordinate RMSD using all heavy atoms. The model whose RMSD is smallest to the others were selected as the representative one.

5. *Minor point: Line 48, 50: 200 kDa will be 200 Da. 1000 kDa will be 1000 Da.*

Response: We have revised the manuscript accordingly.

Reviewers' Comments:

Reviewer #1:

Remarks to the Author:

I have no further comments.

Reviewer #2:

Remarks to the Author:

I'm entirely pleased with how the reviewers integrated my feedback. The computational method for identifying glycan binding sites is both sturdy and inventive, marking progress toward further optimising ligands and refining them in order to optimise drug-design process.

The methodology presented stands on par with other existing approaches, surpassing them in performance. Additionally, the coder demonstrates exceptional robustness when compared to the established gold standards in the field.

The authors conducted a thorough analysis to tackle potential flaws in their methodology, effectively addressing any associated limitations.

Based on the previous review and above mentioned reasons, I endorse this work for publication, and I also plan to implement the proposed methodology in my own systems.

Reviewer #3:

Remarks to the Author:

No additional comments. The paper is now revised properly.